# Survival of cyanobacteria and mitigation of Fe(II) toxicity effects in a silica-rich Archean ocean

Carolin L. Dreher [1], Olaf A. Cirpka[2], Manuel Schad [3,4], Kurt O. Konhauser [3] & Andreas Kappler [1,5] ✉

Banded iron formations (BIF) were deposited abundantly between 2.7-2.4 Ga from iron- and silica-rich oceans, with cyanobacterial oxygen ($O_2$) as a possible oxidant for $Fe(II)_{(aq)}$ oxidation and Fe(III) oxyhydroxide precipitation. However, toxic reactive oxygen species (ROS) from $Fe(II)/O_2$ interactions might have inhibited cyanobacterial growth, contributing to the delay between cyanobacterial evolution (>3.0 Ga) and the Great Oxidation Event (2.5 Ga). Here, we explored the impact of $Fe(II)_{(aq)}$ and $SiO_{2(aq)}$ on *Synechococcus sp.* PCC 7002. High $Fe(II)_{(aq)}$ ($> 500\,\mu M$) increased ROS formation, but elevated $SiO_{2(aq)}$ ($2200\,\mu M$) suppressed ROS formation, promoting growth and $O_2$ production. Diel light cycles further reduced ROS formation compared to continuous illumination. Modelling $O_2$ distribution based on experimental rates revealed oxygenated surface waters at relevant upwelling rates. Together, our results indicate that high $SiO_{2(aq)}$ and day-night-light cycles in Archean oceans mitigated ROS stress, enabling cyanobacterial proliferation and enhancing their role in Earth's oxygenation and BIF deposition.

Sometime between the evolution of oxygen ($O_2$)-producing cyanobacteria, possibly as early as 3.4 billion years ago[1,2] and the permanent rise of atmospheric oxygen during the Great Oxidation Event (GOE) at 2.5–2.3 Ga[3], free $O_2$ began to accumulate in seawater. This accumulation likely began in the form of localized and/or transient oxygen oases, with estimated $O_2$ concentrations ranging from 5 to 100 μM[4–7], before episodically expanding across broader regions of the continental shelf[8–10]. The reasons for the protracted delay between the emergence of cyanobacteria and the GOE remain unresolved but may, at least in part, reflect environmental constraints that inhibited cyanobacterial growth and dispersal[11,12].

One of the most distinctive sedimentary archives of the Archean, BIFs, may hold clues to these limiting environmental conditions (see[13] for review). These chemical sediments consist predominantly of iron- (15–40 wt.% Fe) and silica-rich (40–60 wt.% $SiO_2$) layers that

precipitated from seawater following the oxidation of $Fe(II)_{(aq)}$, with estimated concentrations ranging from 0.03 to 0.5 mM[14,15], and possibly exceeding 1.0 mM in some settings[16,17]. Prior to the emergence of silica-precipitating microorganisms (e.g., diatoms), silica inputs from chemical weathering of early continental crust and hydrothermal vents led to elevated dissolved silica concentrations in the seawater. These concentrations were constrained chiefly by saturation with respect to cristobalite or amorphous silica, on the order of 0.67-2.2 mM[18,19]. Recent studies further suggest that under these 'high silica' concentrations, the primary precipitate may have been a gel-like composite of ferric oxyhydroxide and silica[20–22].

While the deposition of the oldest BIF was probably linked to anoxygenic phototrophic Fe(II)-oxidizers[23,24], photochemical Fe(II) oxidation[25,26] or abiotic deposition processes such as chemical greenalite deposition[27,28], the proliferation of cyanobacteria would have

[1]Geomicrobiology, Department of Geosciences, University of Tuebingen, Tuebingen, Germany. [2]Hydrogeology, Department of Geosciences, University of Tuebingen, Tuebingen, Germany. [3]Department of Earth and Atmospheric Sciences, University of Alberta, Edmonton, Alberta, Canada. [4]Now: GFZ Helmholtz Centre for Geosciences, Section of Geomicrobiology, Potsdam, Germany. [5]Cluster of Excellence EXC 2124, Controlling Microbes to Fight Infection, University of Tuebingen, Tuebingen, Germany. ✉e-mail: andreas.kappler@uni-tuebingen.de

accelerated Fe(II) oxidation and thus BIF deposition[13,29]. Cyanobacteria fix and reduce $CO_2$ using water as electron donor to form biomass, also yielding $O_2$ as by-product (reaction 1). As such, they were key players in the oxygenation of Earth[30,31].

$$6CO_2 + 12H_2O \xrightarrow{h\nu} C_6H_{12}O_6 + 6O_2 + 6H_2O \qquad (1)$$

In early Precambrian oceans, the $O_2$ may have reacted with $Fe(II)_{(aq)}$ to form $Fe(III)_{(aq)}$, which hydrolyzed into Fe(III) oxyhydroxide minerals. The reaction between dissolved $Fe(II)_{(aq)}$ and $O_2$ may also yield reactive oxygen species (ROS), that is, highly reactive oxygen-containing molecules (i.e., the Fenton reaction[32]):

$$O_2 + Fe^{2+} \rightarrow Fe^{3+} + O_2^- \bullet \qquad (2)$$

$$O_2^- \bullet + Fe^{2+} + 2H^+ \rightarrow Fe^{3+} + H_2O_2 \qquad (3)$$

$$Fe^{2+} + H_2O_2 \rightarrow Fe^{3+} + OH^- + OH\bullet \qquad (4)$$

Briefly, superoxide ($O_2^- \bullet$), generated by the reaction of $O_2$ and $Fe(II)_{(aq)}$ (reaction 2; note we provide actual chemical species here versus the generic Fe(II)), further reacts with $Fe(II)_{aq}$ to produce $Fe(III)_{(aq)}$ and hydrogen peroxide ($H_2O_2$) (reaction 3). The $H_2O_2$ then oxidizes the remaining $Fe(II)_{(aq)}$ to $Fe(III)_{(aq)}$, along with the formation of hydroxide anions ($OH^-$) and hydroxyl radicals (OH•) (reaction 4). While the $Fe^{3+}$ drives the formation of Fe(III) minerals, the generated radicals can undergo further reactions, forming additional ROS like ozone ($O_3$) and hydroperoxyl radicals (OOH•) (for a review see ref. 33).

ROS are harmful to cells; extracellular ROS can damage cell membranes through oxidative stress[34], while intracellular ROS can oxidize critical biomolecules such as RNA, DNA, proteins, and lipids[35]. Therefore, ROS formation in Archean oceans may have profoundly influenced microbial activity and evolution. Previously, Swanner and colleagues investigated ROS production under early ocean analog conditions using phosphate-buffered saline (PBS) media and varying concentrations of $Fe(II)_{(aq)}$ (0.1 and 1 mM) in the presence of *Synechococcus sp.* PCC 7002 at a density of $2 \times 10^7$ cells/mL[12]. Under continuous light, they found a 4-fold increase in intracellular ROS levels in the presence of 1 mM $Fe(II)_{(aq)}$ compared to 0.1 mM $Fe(II)_{(aq)}$, suggesting potential toxicity to cyanobacteria at elevated $Fe(II)_{(aq)}$ concentrations. Additionally, in incubation experiments of cyanobacteria under initially anoxic conditions (in the presence of up to 300 μM $Fe(II)_{aq}$) reduced autofluorescence of the cyanobacteria was reported, accompanied by significantly lower actinic yield, and decreased growth rates at $Fe(II)_{(aq)}$ concentrations >180 μM.

In a follow-up study, Swanner and colleagues conducted Fe(II) oxidation experiments with *Synechococcus* sp. PCC 7002 under continuous illumination and a range of $Fe(II)_{(aq)}$ concentrations (7.5, 29, 577, and 4805 μM)[11]. At lower $Fe(II)_{aq}$ concentrations, cultures changed from colorless to green, indicating a high density of cyanobacterial cells. By contrast, higher $Fe(II)_{(aq)}$ concentrations produced an orange coloration, consistent with the formation of Fe(III) oxyhydroxide minerals. These observations were supported by direct cell counts, which confirmed reduced growth at higher $Fe(II)_{(aq)}$ concentrations, again suggesting toxic effects. Microscopic analyses further revealed that cells grown under high $Fe(II)_{(aq)}$ concentrations were smaller and exhibited reduced pigment content, as indicated by lower carotenoid and chlorophyll concentrations inferred from optical density measurements. Although Fe(II) oxidation rates increased with rising $Fe(II)_{(aq)}$ concentrations (from 13 μM/day at 29 μM to 475 μM/day at 4805 μM) the rate of $O_2$ production in the liquid phase decreased from 120 μM/day to 47 μM/day, indicating diminished cellular activity. Notably, $O_2$ was already detectable during Fe(II) oxidation at 29 and 475 μM $Fe(II)_{(aq)}$ but absent at 4805 μM $Fe(II)_{(aq)}$. Taken together, these findings suggest that elevated $Fe(II)_{(aq)}$ concentrations in Archean seawater could have imposed oxidative stress, limiting cyanobacterial productivity and expansion. This could potentially explain the lag between the evolution of cyanobacteria and the GOE[11,12].

However, the potential mitigating impact of high $SiO_{2(aq)}$ concentrations on $Fe(II)_{(aq)}$ toxicity has not been previously considered. Our working hypothesis is that silica binds $Fe(II)_{aq}$, forming Fe-Si-aggregates that slow Fe(II) oxidation and consequently inhibit ROS formation. Previous studies have shown that reduced Fe(II) oxidation rates lead to lower ROS formation[36]. Consistent with this, Dreher and colleagues[37] showed, using SEM/EDS, that Fe-Si aggregates precipitate under similar experimental conditions, further supporting our hypothesis. To test this hypothesis, we experimentally examined the combined effects of $Fe(II)_{(aq)}$ and $SiO_{2(aq)}$ on cyanobacterial $O_2$ production, Fe(II) oxidation, and potential ROS-related toxicity. Specifically, we incubated *Synechococcus* sp. PCC 7002 under alternating day-night-cycles (16 h light; 8 h dark) in artificial seawater medium, in the absence of $SiO_{2(aq)}$ (0 μM, which we refer to as 'no-silica') and in the presence of $SiO_{2(aq)}$ (2200 μM, which we refer to as 'high-silica') and varying initial $Fe(II)_{(aq)}$ concentrations (0, 500, 2500, 5000 μM). Over the course of the experiments, we monitored $O_2$ production, $Fe(II)_{aq}$ and $SiO_{2aq}$ concentrations, total iron ($Fe_{(tot)}$), cell numbers, and ROS production to assess the interdependent effects of $Fe(II)_{(aq)}$ and $SiO_{2(aq)}$ on cyanobacterial physiology and oxidative stress.

## Results

### Effects of different silica concentrations on Fe(II) oxidation by cyanobacterial $O_2$ under alternating day-night-cycles

We conducted Fe(II) oxidation experiments, under initially anoxic conditions, using $Fe(II)_{(aq)}$ concentrations ranging from 0.5 to 5 mM. Following inoculation with the cyanobacterial strain *Synechococcus* sp. PCC 7002, we monitored Fe(II), Fe(tot), cyanobacterially produced $O_2$, dissolved Si, and cell counts over time (for sterile controls see Fig. S3). The results are plotted in Fig. 1 on the respective timescale of the active Fe(II) oxidation, as this sets the focus on the period when there was still $Fe(II)_{(aq)}$ in the system, which is the most relevant for determining toxicity effects and the latter modeling. In setups without amendment of additional $Fe(II)_{(aq)}$, all cultures exhibited a color change from clear to green within two days, regardless of $SiO_{2(aq)}$ concentration (Fig. 1C), indicating cyanobacterial growth. In the 'high silica' setups compared to the 'no silica' setups, cell growth was initially more rapid but reached a slightly lower maximum cell density (Fig. 1C, Table 1). Under both experimental conditions, $O_2$ concentrations peaked at 500 μM after 60 days (Fig. 1B). In the setups without additional Fe(II) amendment, the $Fe(II)_{(aq)}$ concentration remained relatively stable throughout the experiments, averaging approximately 50 μM in with and without both silica setups (Fig. 1A), a level attributable to the $Fe(II)_{(aq)}$ added as nutrient in the medium. In the 'high silica' setups, the initial $SiO_{2(aq)}$ was ca. 1600 μM (Fig. S1A). However, after 30 days, solid silica precipitation started, ultimately reducing the $SiO_{2(aq)}$ concentration to several hundred μM by the end of the experiment.

In experiments with amendment of 500 μM of $Fe(II)_{(aq)}$, all bottles turned green within two days, regardless of the $SiO_{2(aq)}$ concentration (Fig. 1F). Growth curves were similar across setups, with no discernible effect of $SiO_{2(aq)}$ (Fig. 1F, Fig. 2, Table 1), and maximum cell densities were reached after 87 days (Fig. 2). In both 'no silica' and 'high silica' setups, $O_2$ accumulated in the headspace following complete Fe(II) oxidation, peaking at 450 μM after 60 days (Fig. 1E).

Within the first five days of the experiment, 80–90% of the added 500 μM $Fe(II)_{(aq)}$ was oxidized in both 'no silica' and 'high silica' setups (Fig. 1D). With and without silica, the Fe(II) oxidation rates were similar within the calculated error (Table 1). In the 'no silica' and 'high silica' setups, Fe(II) oxidation rates were similar at about 100 μM/day (Fig. 1D; Table 1). In the 'high silica' setups (Fig. 1B), silica precipitation began after 60 days, reducing $SiO_{2(aq)}$ to approximately 100 μM.

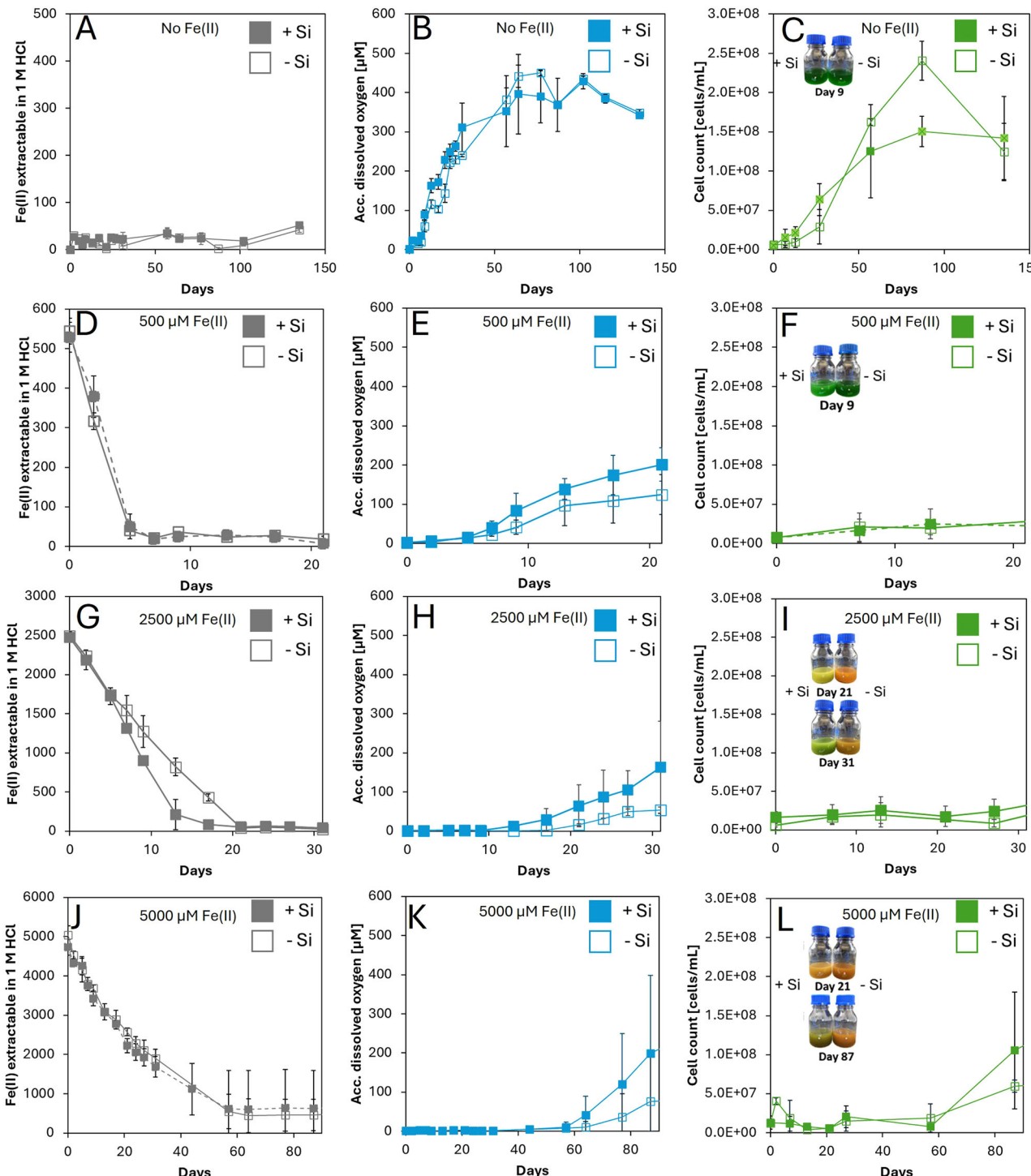

**Fig. 1 | Geochemical data of Fe(II) oxidation experiments containing *Synechococcus* sp. *PCC 7002*.** Panels **A**–**C** show the results of setups without amendment of Fe(II)$_{(aq)}$. Panels **D**–**F** show the results of setups with amendment of 500 μM Fe(II)$_{(aq)}$, panels **G**–**I** with 2500 μM Fe(II)$_{(aq)}$, and panels **J**–**L** with 5000 μM Fe(II)$_{(aq)}$. Filled symbols indicate setups with 2200 μM silica ('high silica'), while empty symbols indicate setups without silica ('no silica'). Fe(II) (gray), accumulated oxygen (blue), cell numbers (green) are plotted as average values from triplicates with the standard deviations shown as error bars. Please note that the time scale (x-axis) varies for the different plots because of the differences in the duration of active Fe(II) oxidation.

In experiments amended with 2500 μM of Fe(II)$_{(aq)}$, the bottles turned orange in the absence of added silica (Fig. 1I), contrasting with experiments with 0 and 500 μM added Fe(II)$_{(aq)}$. However, in the presence of silica, the 2500 μM Fe(II) bottles turned yellow by day 21 and green by day 31. The orange coloration in the 'no silica' setup suggests abiotic Fe(II) oxidation and the formation of Fe(III) oxyhydroxide minerals, likely caused by cyanobacterially produced O$_2$, but with limited cell growth and metabolic activity. By contrast, the green coloration in the 'high silica' setups indicates extensive cyanobacterial proliferation. It is important to interpret these color changes cautiously, as higher Fe(II) concentrations can intensify the orange hue regardless of biological activity, potentially confounding visual comparisons between setups.

**Table 1 | Geochemical data of the Fe(II) oxidation (FeOx) experiments**

| Silica [µM] | Fe(II) [µM] | FeOx rate µM/day | Length of FeOx [days] | Maximum oxygen conc. [µM] | Average maximum cell count [cells/mL] | Green liquid observed after x days |
|---|---|---|---|---|---|---|
| 0 | 0 | – | – | 450 after 60 days | $2.5 \times 10^8$ after 87 days | 2 |
| 2200 | 0 | – | – | 450 after 60 days | $1.7 \times 10^8$ after 87 days | 2 |
| 0 | 500 | $101 \pm 8$ | 5 | 450 after 60 days | $2.0 \times 10^8$ after 87 days | 2 |
| 2200 | 500 | $96 \pm 10$ | 5 | 450 after 60 days | $2.0 \times 10^8$ after 87 days | 2 |
| 0 | 2500 | $129 \pm 5$ | 21 | 350 after 100 days | $2.3 \times 10^8$ after 140 days | - |
| 2200 | 2500 | $174 \pm 11$ | 17 | 490 after 100 days | $1.7 \times 10^8$ after 87 days | 31 |
| 0 | 5000 | $77 \pm 6$ | 58 | 290 after 140 days | $0.6 \times 10^8$ after 140 days | – |
| 2200 | 5000 | $72 \pm 20$ | 58 | 100 after 140 days | $2 \times 10^8$ after 140 days | 87 |

The FeOx rate [µM/day] of the triplicates with the standard deviation, the length of active FeOx [days], the average maximum accumulated oxygen concentration [µM], the average maximum cell count [cells/mL,] and the amount of days after which the liquid turned green of the setups containing 0 µM added silica ('no silica') and 2200 µM added silica ('high silica') with amendment of 0–5000 µM Fe(II)$_{(aq)}$.

Cell counts revealed that in the 'no silica' setups, cyanobacterial growth was delayed, with cell numbers increasing only after 25 days (Fig. 1I). Conversely, cell proliferation in the 'high silica' setups commenced immediately after inoculation and ultimately reached similar densities by day 150 (Fig. 1I). Silica availability had a marked influence on O$_2$ accumulation. In the 'no silica' setups (Fig. 1H), O$_2$ accumulated only after day 20 (ca. 50 µM by day 31). In comparison, in the 'high silica' setups (Fig. 1H) O$_2$ accumulated as early as day 10, reaching 150 µM by day 31.

In the 'no silica' setups, the Fe(II) oxidation rate was lower compared to the 'high silica' setups (Fig. 1G; Table 1). Initial SiO$_{2(aq)}$ concentration in the 'high silica' setups were 1500 µM (Fig. S1C, Fig. 1C) and decreased to 700 µM within the first 20 days. By day 90, variability emerged among the triplicate bottles: one replicate maintained a stable SiO$_{2(aq)}$ concentration of 700 µM, whereas the other two showed continued declines to 300 µM and 100 µM, respectively.

In experiments amended with 5000 µM of Fe(II)$_{(aq)}$, the 'no silica' setups remained orange until day 87 (Fig. 1L), indicating Fe(III) oxyhydroxide formation and limited cyanobacterial growth. By contrast, the 'high silica' setups turned yellow by day 21 and green by day 87, reflecting more substantial cyanobacterial growth. Consistent with these color changes, total cell numbers were higher in the 'high silica' setups. In the 'no silica' setups, cell growth was delayed until Fe(II) oxidation was complete, after which cell densities increased to $5 \times 10^7$ cells/mL (Fig. 1L). In one replicate, cyanobacteria eventually recovered, reaching $1.5 \times 10^8$ cells/mL by day 140 Fig. 2. In the 'high silica' setups, final cell densities varied among the three replicates, reaching $2.5 \times 10^8$, $1.5 \times 10^8$, and $5 \times 10^7$ cells/mL, respectively (Fig. 1L).

O$_2$ accumulated in the 'no silica' setups after 60 days, reaching ca. 80 µM by day 87 (Fig. 1K), although replicate variability was high: one bottle showed no detectable O$_2$, while another reached 300 µM O$_2$ by day 140. In contrast, O$_2$ concentrations in the 'high silica' setup increased after 57 days (Fig. 1K) and reached 200 µM by day 87. Two out of three replicates further reached 400 µM O$_2$ by day 100, whereas in the third, in which only 60% of the Fe(II) was oxidized, showed no detectable O$_2$.

The Fe(II) oxidation rates were similar between the 'no silica' and 'high silica' setups (Fig. 1J; Table 1), but were notably lower than the rates observed in experiments with 2500 µM Fe(II)$_{(aq)}$. In the 'high silica' setups, the initial SiO$_{2(aq)}$ concentration of 1500 µM declined to 500 µM by day 31 (Fig. S1D), after which it either stabilized or continued to precipitate, reaching final concentrations between 100 and 200 µM.

**Improvement of long-term cyanobacterial cell viability by silica**
In addition to experiments focusing on cyanobacterial growth and activity i.e., cell numbers, O$_2$ production, resulting Fe(II) oxidation and dissolved silica (see Fig. S1), we explored the role of silica in promoting long-term cell viability experiments by observing potential mitigation effects of the harmful effects of ROS generated in the presence of both Fe(II)$_{(aq)}$ and O$_2$. In all setups, the presence of Fe(II)$_{(aq)}$ delayed both cell growth and O$_2$ production (Fig. 2). In the 500 µM Fe(II)$_{(aq)}$ (Fig. 2 E), 'high silica' setups initially showed higher O$_2$ accumulation during the first 70 days, with both setups converging at 350 µM by day 60. In the 2500 µM Fe(II) experiments (Fig. 2G–I), the presence of SiO$_{2(aq)}$ led to significantly enhanced cyanobacterial activity: by day 150, O$_2$ concentrations reached 500 µM and cell densities ca. $2.4 \times 10^8$ cells/mL, compared to 350 µM O$_2$ and $1.8 \times 10^8$ cells/mL in the absence of SiO$_{2(aq)}$. The beneficial effect of silica was even more pronounced in the 5000 µM Fe(II) setups. After 250 days, O$_2$ accumulation in the 'high silica' setups reached 250 µM, with cell densities of $2 \times 10^8$ cells/mL, while the 'no silica' setups showed limited cell growth, reaching only $5 \times 10^7$ cells/mL and 100 µM O$_2$.

**Formation of ROS in incubations of strain PCC 7002 with Fe(II)**
Based on the lower cell growth and Fe(II) oxidation rates at higher Fe(II)$_{(aq)}$ concentrations (500, 2500–5000 µM), we hypothesized that cyanobacterially produced O$_2$ reacted with Fe(II)$_{(aq)}$ to generate ROS, potentially leading to cellular stress or toxicity. However, our results also indicated that SiO$_{2(aq)}$ mitigates these negative effects, as evidenced by higher cell densities, earlier O$_2$ accumulation, and increased Fe(II) oxidation rates in silica-amended setups. To further evaluate the impact of varying Fe(II)$_{(aq)}$ and SiO$_{2(aq)}$ concentrations on ROS production, we subsequently constructed a series of experiments using *Synechococcus* sp. PCC 7002 cell suspensions ($5 \times 10^9$ cells/mL) exposed to different Fe(II)$_{(aq)}$ concentrations (0, 500, 2500 or 5000 µM), both in the absence and presence of 2200 µM SiO$_{2(aq)}$.

Our results (Fig. 3, Fig. S2, Table S1) showed that in the absence of Fe(II)$_{(aq)}$, ROS fluorescence signals were generally very low, yet slightly elevated in the presence of SiO$_{2(aq)}$ compared to setups without SiO$_{2(aq)}$ (ca. 500 a.u. and 400 a.u., respectively) (Fig. 3). After the amendment of 500 µM Fe(II)$_{(aq)}$, the ROS fluorescence signal, with and without silica, was similar to the setups without Fe(II), suggesting no significant ROS formation at this Fe(II)$_{(aq)}$ concentration. However, in the presence of 2500 µM of Fe(II)$_{(aq)}$, we observed a pronounced increase in ROS fluorescence in the absence of silica (2000 a.u.), while the signal remained near baseline in silica-amended setups. A similar trend was observed at 5000 µM Fe(II)$_{(aq)}$, where ROS levels reached 1300 a.u. without silica, but remained negligible in the presence of silica (Fig. 3). These results suggest that high Fe(II) concentrations promote ROS generation, and that dissolved silica effectively suppresses ROS formation under these conditions.

**Numerical modeling of the experimental O$_2$ data**
In order to simulate the total O$_2$ production, we incorporated our raw laboratory data into a numerical model to simulate the total O$_2$

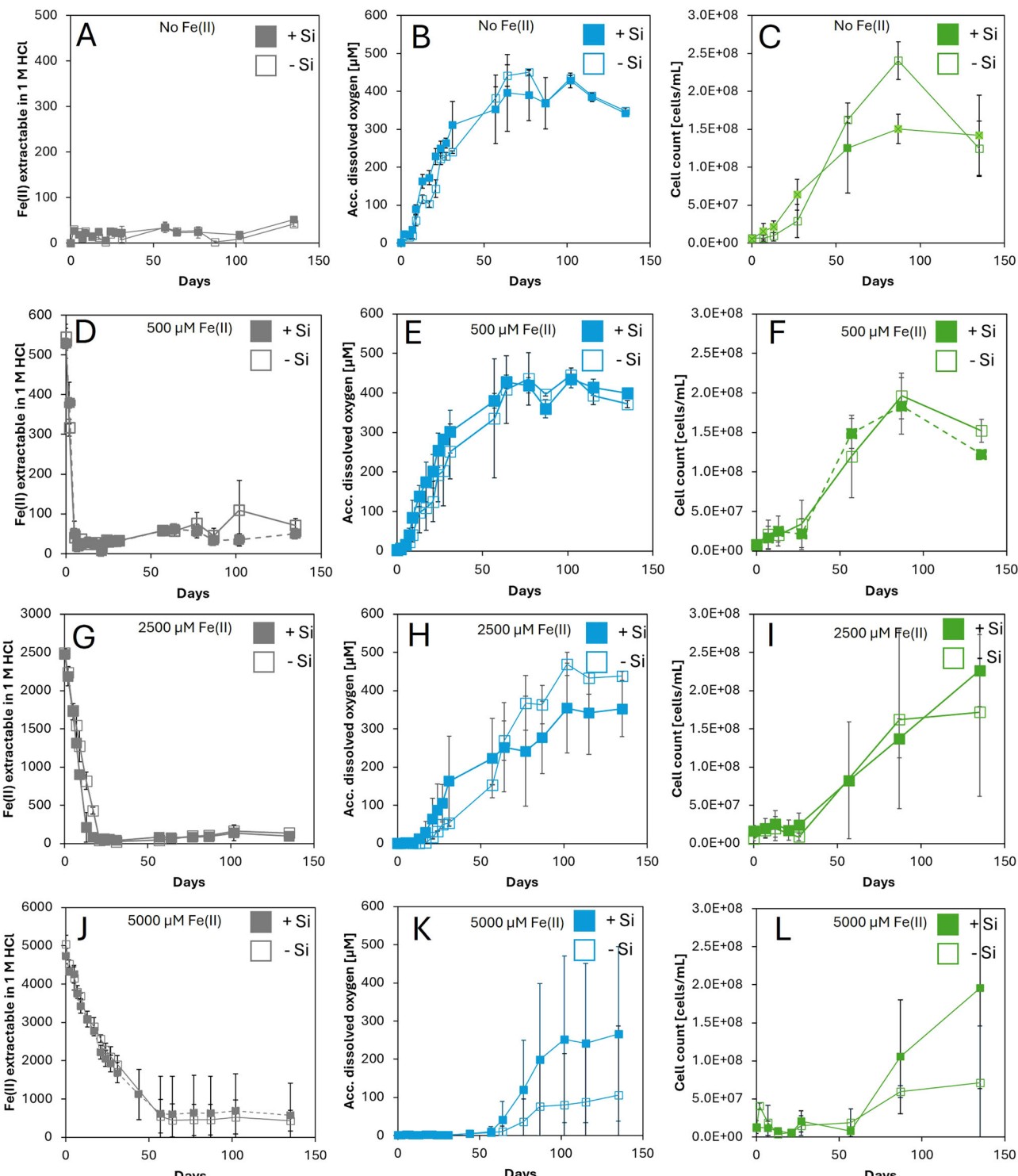

**Fig. 2 | Geochemical data of Fe(II) oxidation experiments containing *Synechococcus* sp. PCC 7002 over the entire course of the experiment.** Panels **A**–**C** show the results of setups without amendment of Fe(II)$_{(aq)}$. Panels **D**–**F** show the results of setups with amendment of 500 μM Fe(II)$_{(aq)}$, panels **G**–**I** with 2500 μM Fe(II)$_{(aq)}$, and panels **J**–**L** with 5000 μM Fe(II)$_{(aq)}$. Filled symbols indicate setups with 2200 μM silica ('high silica'), while empty symbols indicate setups without silica ('no silica'). Fe(II) (gray), accumulated oxygen (blue), cell numbers (green) are plotted as average values from triplicates with the standard deviations shown as error bars.

production. To account for the observed plateau of O$_2$ concentration at approximately 450 μM (Fig. 4A–E), we differentiated between 'active' cells (O$_2$-producing; green solid line) and 'inactive' cells (non–O$_2$-producing; calculated as the difference between total cells and measured active cells). The 'inactive' group included both dead cells and those estimated to have received insufficient light due to increased cell density during the experiment. While cultures were

initially colorless, they gradually developed an intense dark green coloration, increasing turbidity, and thereby reducing the average light intensity available to each bacterial cell.

We also calculated the total O$_2$ production in our experiments, accounting for both the dissolved O$_2$ in the liquid and free O$_2$ present in the headspace. Based on the equilibrium relationship between the aqueous solution and headspace (see Supplementary Information,

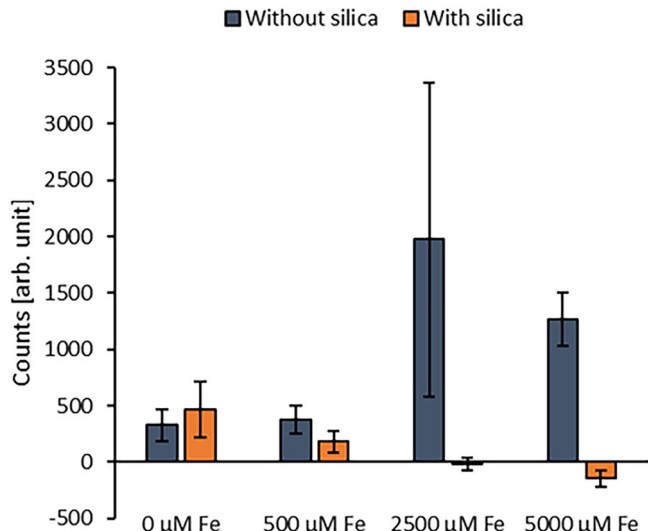

**Fig. 3 | Impact of silica on reactive oxygen species (ROS) formation at different iron concentrations.** Average ROS fluorescence signals relative to an anoxic, abiotic control containing 5000 μM Fe(II)$_{(aq)}$ with and without 2200 μM SiO$_{2(aq)}$. The experimental setups contained strain PCC 7002 cells, either 0, 500, 2500, or 5000 μM Fe(II)$_{(aq)}$ in the absence (blue) and presence of 2200 μM SiO$_{2(aq)}$ (orange). Please note that the normalization of all measurements to the abiotic control can result in negative values. Error bars show the standard deviation calculated from quadruplicate setups.

formula 9), we estimate that O$_2$ production (including the headspace) is approximately 42 times greater than measurements based solely on dissolved O$_2$. This correction is crucial for accurately quantifying true O$_2$ production in our experimental setup.

### Oxygen and Fe(II) distribution modeled for the ancient water column

To scale our laboratory-derived O$_2$ production to early ocean conditions, we calculated the oxygen production per cell ($r_{photo}$) from the first two days of the experiment, a period when no detectable O$_2$ accumulated in the liquid phase and no measurable cell growth occurred (see "Methods" for details). We calculated the oxygen and Fe(II)$_{aq}$ distribution for three upwelling rates: 4, 95, and 473 m/yr. 4 m/yr represents the average upwelling velocity of modern oceans, while 95 m/yr and 473 m/yr are modeled average and enhanced oceanic upwelling velocities in the mid-Cretaceous North Atlantic Ocean, respectively[38]. Using an upwelling rate of 4 m yr$^{-1}$ (Fig. 5A–D), our model predicted O$_2$ concentrations of at least 400 μM between 50 and 100 m depth in the water column, with the potential for substantially higher values. Given that the maximum O$_2$ solubility in an open systems at room temperature and atmospheric pressure is ~300 μM[39,40] (higher in closed systems), these results indicate that such upwelling rates would saturate and fully oxidize the photic zone, leading to complete oxidation of Fe(II)$_{(aq)}$ and excess O$_2$ outgassing to the atmosphere.

By increasing the upwelling rate to 95 m yr$^{-1}$, our model predicts that, at a cyanobacterial abundance of 10$^5$ cells mL$^{-1}$ (Fig. 5E, G), Fe(II)$_{(aq)}$ would have reached the ocean surface. Oxygen concentrations at the surface would have reached between 0.04 and 0.4 μM maximum concentration. At a higher theoretical abundance of 10$^6$ cells mL$^{-1}$ (Fig. 5F, H), however, the photic zone remained Fe(II)$_{aq}$-free down to depths of ~200 m or ~125 m, respectively. In these high-cell-density scenarios, O$_2$ accumulated to much higher concentrations within the upper 0–50 m of the water column, with modeled values reaching (200 μM) or exceeding (800 μM) maximum O$_2$ saturation.

At an upwelling rate of 473 m/yr, Fe(II)$_{(aq)}$ reached the water surface in most scenarios (Fig. 5I, K, L). Under these conditions, O$_2$

accumulation at the top of the water column was limited, ranging from 0.03 μM (Fig. 5K) to 0.8 μM (Fig. 5L). In scenario J, however, Fe(II)$_{(aq)}$ was absent only from the upper 50 m, allowing O$_2$ to accumulate to 80 μM near the water surface. Overall, the results highlight the strong control of upwelling rate on O$_2$ and Fe(II)$_{(aq)}$ dynamics. At low upwelling rates (4 m/yr), the photic zone was fully oxidized, whereas at 96 m/yr, O$_2$ accumulated only near the surface. In contrast, most scenarios with rapid upwelling (473 m/yr) resulted in Fe(II)$_{(aq)}$ reaching the surface, suppressing O$_2$ buildup and preventing full saturation.

## Discussion

### ROS formation and the mechanistic function of Si for ROS prevention

Our results demonstrate a clear relationship between Fe(II)$_{(aq)}$ concentrations and the extent of ROS formation in the absence of SiO$_{2(aq)}$, generally supporting the findings of Swanner and colleagues[11,12]. Differences between our study and theirs can likely be attributed to variations in experimental setups, such as constant illumination versus day-night-cycles, and static versus agitated conditions (see above). Additionally, minor methodological variations, including the use of phosphate-buffered saline (PBS)[12] versus TRIS buffer in our study, as well as differences in fluorometric detection systems, may have contributed to the observed differences. Nevertheless, despite the variations, our study demonstrates a protective effect of SiO$_{2(aq)}$ against ROS-induced stress in cyanobacteria under conditions relevant to early Precambrian oceans. Even at high Fe(II)$_{(aq)}$ concentrations (2500 and 5000 μM), ROS levels remained below those observed in anoxic, abiotic controls when SiO$_{2(aq)}$ was present (Fig. 3). Furthermore, we measured higher O$_2$ immediately following Fe(II) oxidation in the presence of SiO$_{2(aq)}$ at all Fe(II)$_{(aq)}$ concentrations (Fig. 1), suggesting enhanced cyanobacterial activity. Notably, in the 5000 μM Fe(II)$_{(aq)}$ experiments, SiO$_{2(aq)}$ enabled long-term recovery and sustained growth of *Synechococcus sp*. PCC 7002, with cell densities reaching 2×10$^8$ cells/mL after 150 days, compared with only 5 × 10$^7$ cells/mL in the absence of SiO$_{2(aq)}$.

The protective effect of SiO$_{2(aq)}$ against ROS formation can be attributed to the ability of Si to bind Fe(II)$_{(aq)}$, forming Fe(II)-Si complexes[28,37], that reduce Fe(II) reactivity with O$_2$, thereby inhibiting or reducing Fenton-type reactions and lowering ROS production. The formation of amorphous Fe(II)-Si aggregates, as hypothesized in our experiments, correlates well with previous studies simulating Archean ocean conditions (1 mM SiO$_{2aq}$, 1.1 mM Fe(II)$_{(aq)}$, pH of 6.5–7.5[41,42];). This pH range is consistent with both our experimental conditions (pH 7) and estimates for circumneutral pH in early Precambrian oceans[43,44]. While Fe(II) within these aggregates remained susceptible to oxidation by O$_2$, the resulting products likely precipitated as Fe(III)-Si-aggregates on the seafloor rather than as Fe(II)-Si phases ([37]).

### Day-night-cycles reduce ROS formation compared to continuous light and static conditions

Our results showed that Fe(II)$_{(aq)}$ oxidation rates under day-night-cycle incubations were higher than those reported in a previous study using the same setup—including the same cyanobacterial strain (*Synechococcus* sp. PCC 7002), initial cell densities (10$^6$–10$^7$ cells/mL), similar Fe(II)$_{(aq)}$ and SiO$_{2(aq)}$ concentrations, light intensities (300–500 lux), and identical glassware—but conducted under continuous illumination ([37]) (Table 2). In these continuous light experiments, the presence of 2200 μM silica resulted in remarkably higher cell counts and O$_2$ concentrations compared to the silica-free equivalents[37]. The difference in Fe(II) oxidation rates between day-night cycles and continuous illumination was especially pronounced at the lowest Fe(II)$_{(aq)}$ concentration (500 μM) in the absence of SiO$_{2(aq)}$, where Fe(II) oxidation rates reached 113 μM/day under day-night cycles versus only 32 μM/day under continuous light. The only exception occurred in the setup containing 2500 μM Fe(II)$_{(aq)}$ and 2200 μM SiO$_{2(aq)}$, where Fe(II)

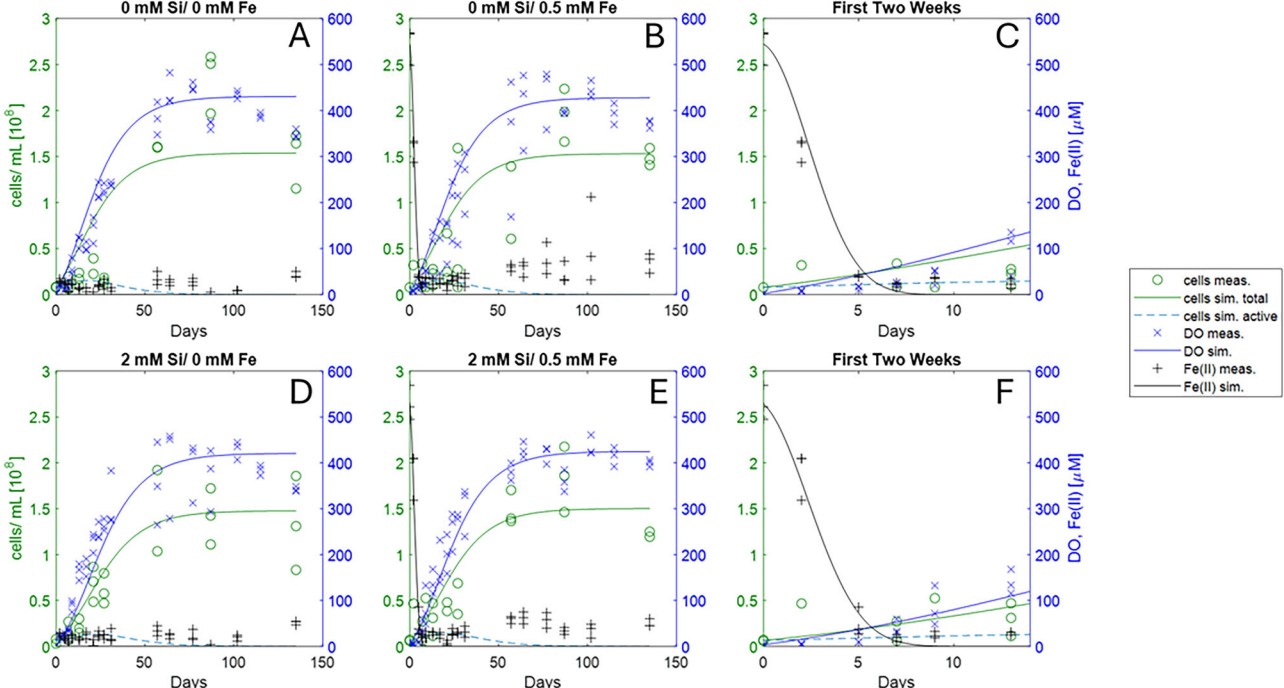

**Fig. 4 | Modeled lab data of the experimental setups containing 500 μM iron without silica (top) and with 2200 μM silica (bottom).** Panel **A** shows the modeled data of setups without amendment of silica or Fe(II)$_{(aq)}$, **B** shows data of setups without silica amendment but with 500 μM Fe(II)$_{(aq)}$. **D** Shows setups with 2200 μM silica without Fe(II)$_{(aq)}$, **E** shows setups with 2200 μM silica and 500 μM Fe(II)$_{(aq)}$. Panels **C** and **F** show the data zoomed into the first 15 days of panels **B** and

**E**, respectively. Green ('cell sim.total'), blue ('DO sim.') and black ('Fe(II) sim.') lines are the modeled total cell densities and oxygen concentrations, respectively, the green circles ('cells meas.'), the blue crosses ('DO meas.') and the black crosses ('FeII) meas.') show the associated raw data points. The bright blue dashed line ('cells sim. active') shows the calculated concentration of active cells. The black solid lines ('Fe(II) sim.') shows the modeled Fe(II) curve.

oxidation rates of 129 and 168 μM/day were observed in day-night-cycles and under continuous illumination, respectively.

The effects of continuous light exposure on phototrophic organisms have been well documented[45], with prolonged and continuous illumination linked to increased ROS formation and cellular damage. In the context of Fe(II) oxidation, Hegler and colleagues examined the influence of light intensity and wavelength under continuous illumination, reporting maximum Fe(II) oxidation rates of 4.5 mmol/L/day at 800 lux, with overall Fe(II) oxidation rates increasing with light intensity[46]. More recently, Nikeleit and colleagues demonstrated that day-night-cycle exposure enhanced Fe(II) oxidation rates in the anoxygenic phototroph *Chlorobium ferrooxidans* strain KoFox, suggesting a physiological benefit of diel variation[47]. Furthermore, day-night-cycles have been shown to stimulate biomass production, protein synthesis, and coenzyme Q10 production in anoxygenic phototrophic bacteria such as *Erythrobacter* sp. NAP1 and *Rhodobacter sphaeroides*[48,49].

During the light periods of day-night-cycles, cyanobacteria produce O$_2$ most efficiently at light intensities between 30 and 50 μmol m$^2$ s$^{-1}$ [50–52]. In dark periods, their metabolic activity is severely limited, with neither O$_2$ nor glucose produced; energy is instead derived from the degradation of internally stored glycogen[53]. Muhetaer and colleagues[52] showed that the effects of continuous light depend on the strain's specific circadian rhythm. For example, *M. aeruginosa* exhibited increased H$_2$O$_2$/ OD$_{730}$ ratios and partial cell death after 8 days of continuous light (e.g., H$_2$O$_2$/ OD$_{730}$ of 400 at 300 μmol m$^{-2}$ s$^{-1}$) compared to two days (H$_2$O$_2$/ OD$_{730}$ of 230 at 300 μmol m$^{-2}$ s$^{-1}$), whereas the strain *P. galeata* showed opposite trends, suggesting significant inter-strain variability in stress responses to prolonged illumination[52]. It is thus plausible that day-night-cycles applied in our experiments were more ideal for the cyanobacteria's circadian rhythm than continuous light exposure. This may explain the enhanced Fe(II) oxidation

rates observed in our experiments compared to previous experiments conducted under continuous illumination, likely due to higher O$_2$ production during the light periods.

In addition to day-night-cycles, shaking the incubations enhanced cyanobacterial growth. Multiple studies have shown that *Synechococcus* strains benefit from fluid movement during growth, as it promotes homogenization of the medium and improves access to nutrients, light, and CO$_2$[54–56]. For example, Kuan and colleagues reported optimal growth rates for *Synechococcus elongatus* PCC7942 at shaking speeds of 100–150 rpm[57]. Our data are consistent with these findings, confirming the fluid agitation—analogous to conditions in the phototrophic zones of ancient oceans—stimulates cyanobacterial activity and, consequently, O$_2$ production.

Beyond shaking, the configuration of incubation bottles also appears to influence cyanobacterial growth. More specifically, experiments with open bottles generally show faster doubling rates and higher cell densities[58] compared to setups using closed bottles[11,12]. This difference may result from continuous headspace ventilation in open bottles, which facilitates O$_2$ removal and potentially reduce ROS formation. However, comparisons between these studies should be made cautiously, as the Fe(II)$_{(aq)}$ concentrations used by Hermann and Gehringer were relatively low (20-120 μM)[58], a range in which Swanner and colleagues also observed minimal ROS-related toxicity[11,12].

### Implications for the oxygen distribution in the seawater

From our laboratory-derived per-cell O$_2$ production rates, we estimate that under low upwelling conditions (4 m yr$^{-1}$) (modern value) and a total Fe(II)$_{(aq)}$ concentration of 529 μM, the entire photic zone should be oxidized at cyanobacterial cell densities of approximately 10$^5$–10$^6$ cells mL$^{-1}$. At substantially higher upwelling rates (95 and 473 m yr$^{-1}$)[38], overall O$_2$ concentrations throughout the water column would be lower; nevertheless, cyanobacterial activity would still

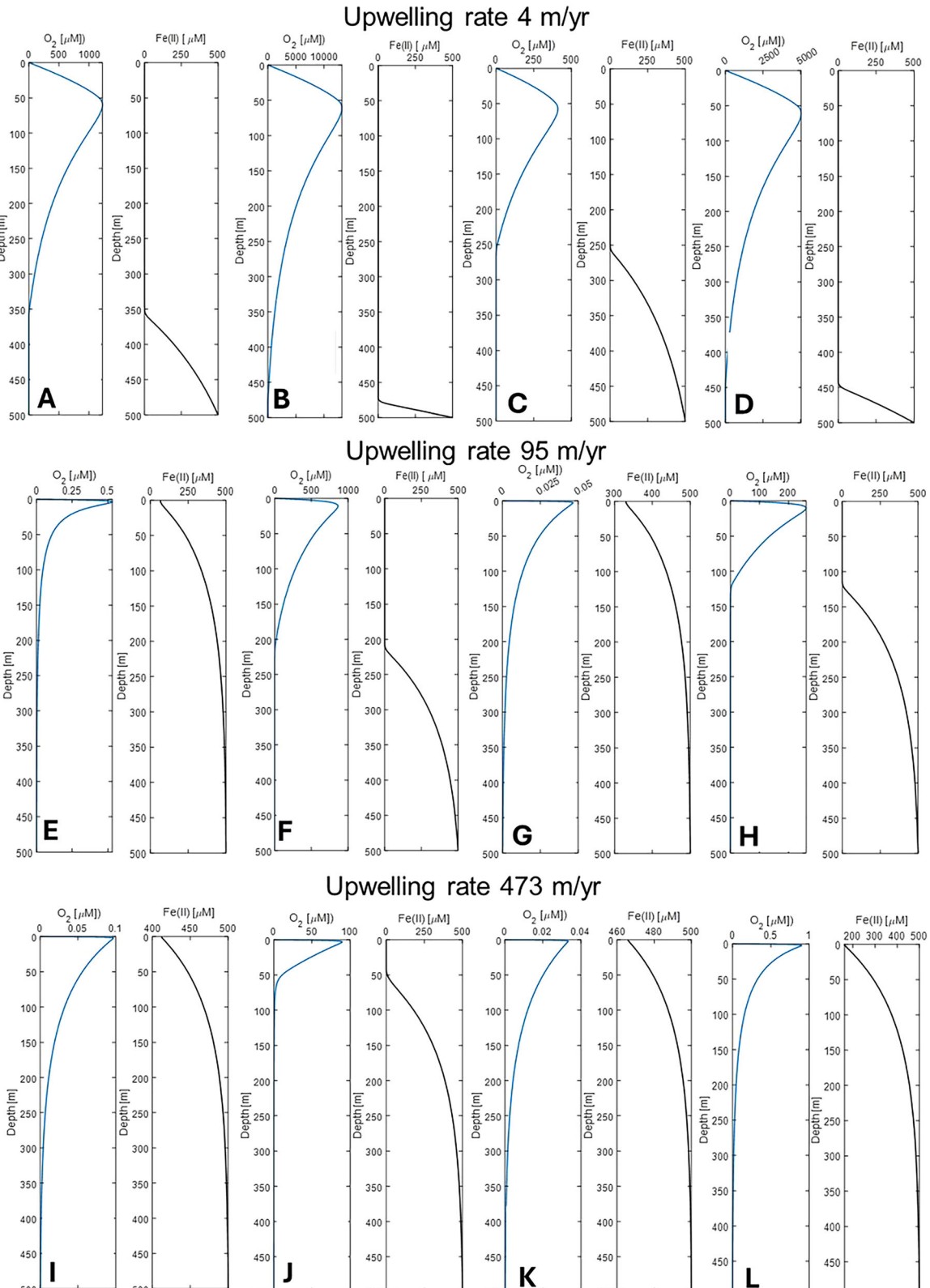

**Fig. 5 | 1D steady-state model of oxygen distribution in the early ocean.** Modeled oxygen (left, blue) and Fe(II) (right, gray) concentrations in the ancient ocean water column for an upwelling velocity of 4 (panels **A**–**D**), 95 (panels **E**–**H**) and 473 m/yr (panels **I**–**L**). Panels **A**, **C**, **E**, **G**, **I**, and **K** are calculated for hypothesized $10^5$ cells/mL in the early ocean, **B**, **D**, **F**, **H**, **J**, **L** for $10^6$ cells/mL. Panels **A**, **B**, **E**, **F**, **I**, and **J** are modeled based on the photosynthetic rate ($r_{photo}$) calculated from the lowest cell count ($6.5 \times 10^5$ cells/mL) we measured in our experiments at d0; **C**, **D**, **G**, **H**, **K**, and **L** are based on the highest cell count ($1.6 \times 10^7$ cells/mL) we measured at d0.

**Table 2 | Fe(II) oxidation rates in day-night-cycles (left column, data from this paper); Fe(II) oxidation rates in continuous light right column, data from Dreher and colleagues[37]**

| Silica [μM] | Fe(II) [μM] | day-night-cycles FeOx rate μM/day | continuous light FeOx rate μM/day |
|---|---|---|---|
| 0 | 500 | 101 ± 7.7 | 32 ± 1 |
| 2200 | 500 | 96 ± 9.7 | 30 ± 1 |
| 0 | 2500 | 129 ± 5.1 | 168 ± 2 |
| 2200 | 2500 | 174 ± 10.8 | 147 ± 15 |
| 0 | 5000 | 77 ± 6.4 | 63 ± 10 |
| 2200 | 5000 | 72 ± 20.2 | 40 ± 2 |

support $O_2$ accumulation at the surface, forming a distinct oxygenated layer.

Incorporating the upwelling rates and $Fe(II)_{(aq)}$ concentrations applied in the studies of Swanner and colleagues[12] into our models (see Fig. S6–S8), and applying our most conservative $O_2$ production rate ($r_{photo}$ of 0.122 μM/day), yields maximum $O_2$ concentrations at least an order of magnitude greater than previously estimated (33 μM of maximum oxygen compared to 2–4 μM of Swanner and colleagues[12]). Moreover, the modest toxicity effects observed at 529 μM $Fe(II)_{(aq)}$ would be even less pronounced at lower concentrations (25–125 μM), as also reported by Swanner et al.[11,12], suggesting that our model predictions may still underestimate true $O_2$ production.

When applying this approach to specific scenarios, it produces results that diverge markedly from earlier models[12]. For example, with 25 μM $Fe(II)_{(aq)}$ and an upwelling rate of 95 m yr$^{-1}$, our model predicts $O_2$ concentrations up to 30 μM and a $Fe(II)_{(aq)}$-free zone extending to more than 200 m depth, whereas the earlier model estimated a maximum of 1–2 μM $O_2$ and a $Fe(II)_{(aq)}$-free zone less than 50 m deep. Under 25 μM $Fe(II)_{(aq)}$ and rapid upwelling (473 m yr$^{-1}$), both models predict $Fe(II)_{(aq)}$ transport to the surface, but our simulations still yield 2.5 μM $O_2$ at the surface, while the earlier study predicted no measurable $O_2$ accumulation. Finally, under low upwelling (4 m yr$^{-1}$) and higher $Fe(II)_{(aq)}$ concentrations (120 μM), our model produces 500 μM $O_2$ with complete $Fe(II)_{(aq)}$ oxidation down to 400 m, in contrast to ~80 μM $O_2$ and oxidation to only 250 m in the earlier model. Collectively, these findings demonstrate that using experimentally derived cyanobacterial $O_2$ production rates results in substantially higher predicted $O_2$ concentrations and deeper $Fe(II)_{(aq)}$-free zones than previous modeling efforts have suggested.

**Oxygenic photosynthesis on early Earth proceeded without harmful ROS stress to cyanobacteria**

Our experiments involving cyanobacterial incubations at high $Fe(II)_{(aq)}$ concentrations in the absence of $SiO_{2(aq)}$ are consistent with previous studies demonstrating that $O_2$-producing cyanobacteria were exposed to ROS under early Earth conditions[11,12]. However, our results suggest that ROS were not necessarily a major physiological stressor for cyanobacteria during this time. Dreher and colleagues[36] reported increasing $Fe(II)_{(aq)}$ toxicity with increasing $Fe(II)_{(aq)}$ concentrations, but observed markedly enhanced bacterial growth in the presence of silica. Based on these results, we hypothesize that silica reacts with $Fe(II)_{(aq)}$, thus distinctly lowering Fe(II) reactivity and preventing Fe(II)-induced ROS formation. First, in experiments with iron and silica concentrations relevant to Archean oceans (500 μM $Fe(II)_{(aq)}$ and 2200 μM $SiO_{2(aq)}$[18,19]), we observed rapid cell growth, high $O_2$ production, and complete $Fe(II)_{(aq)}$ oxidation under simulated day-night-cycles (Fig. 2D–F). Second, we found that $SiO_{2(aq)}$ effectively suppressed ROS formation at $Fe(II)_{(aq)}$ concentrations as high as 5000 μM (Fig. 2G–L and Fig. 3). These findings imply that cyanobacteria inhabiting coastal environments with episodically elevated $Fe(II)_{(aq)}$ concentrations (several mM[8,59]) could have remained viable and active–

particularly in contrast to the lower $Fe(II)_{(aq)}$ concentrations generally inferred for bulk seawater (0.02–500 μM respectively[14,35,60–62]).

The observation that ROS formation was inhibited even at high concentrations of both $SiO_{2(aq)}$ (2200 μM) and $Fe(II)_{(aq)}$ (5000 μM) suggests that these chemical conditions imposed minimal stress to cyanobacteria inhabiting early Precambrian oceans. This finding broadly supports the findings of Swanner and colleagues[11,12], although in our experiments, lower $Fe(II)_{(aq)}$ concentrations did not induce significant ROS formation or toxicity. Taken together, our lab experiments, supported by modeling, indicate that ROS toxicity was likely not a major constraint on cyanobacterial growth or proliferation in early marine environments. Consequently, ROS stress alone is unlikely to explain the temporal lag between the emergence of oxygenic cyanobacteria (> 3 Ga) and the onset of the GOE (2.5–2.3 Ga)[31,63–65].

Our results add to the growing body of evidence demonstrating the geomicrobiological significance of Si-Fe interactions in early Precambrian oceans. For instance, $SiO_{2(aq)}$ binds to reactive surface sites of Fe-bearing minerals such as pyrite ($FeS_2$), stabilizing it against oxidative dissolution and inhibiting ROS formation, a process that might have been relevant in the Archean[66,67]. Indeed, silica coatings on pyrite have been shown to reduce surface-bound sulfate generation during oxidative weathering by up to 91%[68], indicating an increased stability of the pyrite. Furthermore, Fe(III)-silica aggregates can serve as adsorption sites for Fe(II) via outer-sphere complexation[69], promoting scavenging of $Fe(II)_{(aq)}$ and therefore suppressing ROS production. Our results expand this knowledge by demonstrating that $SiO_{2(aq)}$ can inhibit ROS formation directly in the water column by binding to $Fe(II)_{(aq)}$, thereby contributing to a more favorable chemical environment for cyanobacterial activity in early oceans.

## Methods

### Cultivation of microorganisms and growth conditions

The cyanobacterium *Synechococcus* sp. PCC 7002, provided by Gen Enomoto from the University of Freiburg, was cultivated in liquid oxic A+ medium. This medium was prepared using the following stock solutions: Stock 1 (2 mL) (1:2 dilution, containing 3.89 g $FeCl_3 \cdot 6H_2O$, 34.3 g $H_3BO_3$, 4.3 g $MgCl_2 \cdot 4H_2O$, 0.315 g $ZnCl_2$, 0.03 g $MoO_3$, 0.003 g $CuSO_4 \cdot 5H_2O$, and 0.0122 g $CoCl_2 \cdot 6H_2O$ in Milli-Q water to 1 L), stock 2 (10 mL) (100 g Trizma base adjusted to pH 8.2 with HCl in Milli-Q water to 1 L), stock 3 (100 mL) (0.5 g $KH_2PO_4$, 10 g $NaNO_3$, 24.4 g $MgSO_4$, 180 g NaCl, and 6 g KCl in Milli-Q water to 1 L), stock 4 (10 mL) (0.3 g $Na_2EDTA$ in Milli-Q water to 100 mL), stock 5 (10 mL) (2.8 g $CaCl_2$ in Milli-Q water to 100 mL). The PCC 7002 culture was grown on a shaker at 60 rpm to a cell density of approximately $10^8$ cells/mL at 25 °C in an Erlenmeyer flask, under a 40-watt halogen light bulb providing an intensity of 300–500 lux.

### Setup of Fe(II) oxidation experiments

The Fe(II) oxidation experiments aimed to culture the planktonic cyanobacterium *Synechococcus* sp. PCC 7002 in the presence of 0, 500, 2500, and 5000 μM Fe(II), along with 0 and 2200 μM dissolved monomeric silica. These experiments were set up in triplicate bottles each to analyze cell growth and accumulated oxygen concentration after complete Fe(II) oxidation by the $O_2$ produced by the cyanobacteria. The bottles were kept at room temperature on a shaker at 60 rpm with day-night-cycles of 8/16 h at intensities of 300–500 lux. Each experiment was conducted in 250 mL Schott bottles containing 100 mL of liquid medium. Anoxic artificial seawater medium (ASW) was prepared with the following composition: 17.3 g NaCl, 8.6 g $MgCl_2 \cdot 6H_2O$, 0.025 g $MgSO_4 \cdot 7H_2O$, 0.99 g $CaCl_2 \cdot 2H_2O$, 0.39 g KCl, 0.059 g KBr, 0.05 g $KH_2PO_4$, 0.25 g $NH_4Cl$, and 2.5 g $NaHCO_3$ per 1 L of Milli-Q (MQ) water. Salts, the bicarbonate buffer (30 mM), and the monomeric silica stock solution were separately purged with $N_2$ (or $N_2/CO_2$ at 90:10 % for the buffer) and autoclaved. After autoclaving, the salt solution in a Widdel flask was flushed with $N_2/CO_2$ while still

hot (-70–80 °C), and the sterile silica solution was immediately added, resulting in the formation of white flocs. The medium was cooled to 20 °C, the bicarbonate buffer was added, and the pH was adjusted to 7.0 using 1 M anoxic HCl. Due to precipitate formation, the medium was transferred to 2 L Schott bottles and kept at room temperature for two days. It was then filtered through a 0.22 µm PES bottle-top filter inside a glovebox. Vitamins and trace element solutions were added under sterile conditions, and the pH was re-adjusted to 7.0. The prepared medium was aliquoted into sterile 250 mL Schott bottles, with each bottle containing a total volume of 100 mL, which included 5 mL of washed cyanobacteria and the respective $FeCl_2$ solution. During this process, a constant stream of $N_2/CO_2$ was maintained. Oxygen concentrations were measured, and any bottles with more than 3 µM oxygen were flushed with $N_2/CO_2$ again. Subsequently, $FeCl_2$ was added from an anoxic and sterile 100 µM or 1 M stock solution. The pH values were rechecked and remained stable at 7.0. Cyanobacteria were washed three times with ASW before being added to the medium as a 5% inoculum to achieve a final cell density of $10^6$ cells/mL. The experiments were initially fully anoxic, so all oxygen detected during the course of the experiments was produced solely by the inoculated cyanobacteria.

### Quantification of Fe(II) and Fe(III) by the spectrophotometric Ferrozine assay

The ferrozine assay is a spectrophotometric method used to quantify the concentration of Fe(II). Samples were first dissolved and diluted in 1 M HCl to achieve a concentration of up to 1 mM Fe. For Fe(II) analysis, 80 µL of 1 M HCl was pipetted into a 96-well plate, followed by the addition of 20 µL of the sample. After a 15-min incubation, 100 µL of ferrozine solution (0.1 w/v) was added. The mixture was allowed to complex for 5 min before being analyzed using a Multiskan™ GO Microplate Spectrophotometer (Thermo Fisher Scientific, USA). Fe(II) is forming a purple complex with ferrozine molecules, which absorbs light at 562 nm. For total iron ($Fe_{(tot)}$) determination, the procedure was modified by replacing the 1 M HCl with 80 µL of 10% (v/v) hydroxylammonium chloride (HAHCl) before adding the sample. This step was followed by a longer incubation period of 30 min to ensure the complete reduction of Fe(III) to Fe(II). To prevent degradation, both the ferrozine and HAHCl solutions were stored and incubated in the dark, as they are highly light-sensitive. Different standards between 0 and 1000 µM were used to calculate the respective iron concentration.

### Quantification of dissolved silica by the molybdenum blue method

Dissolved silica, including monomeric silica and colloidal silicic acid (Si-Si-colloids), was quantified by the Molybdenum Blue Method[70]. The samples were filtered (0.22 µm, PES) and diluted in MQ to 1 mL. Subsequently, 40 µL of an ammonium heptamolybdate tetrahydrate (6.33 g) together with 50 mL of 4.5 M sulfuric acid solution were added to form a blue complex with the silica. To avoid complexing with phosphate, 40 µL of a 10 g/100 mL MQ oxalic acid solution was added. Lastly, 20 µL of 0.16 M ascorbic acid was added as reductant. After 30–60 min of incubation, the blue silica-molybdenum complexes were quantified at 810 nm. Different standards between 1 and 100 µM were used to calculate the silica concentration. Due to the filtration of the samples (0.22 µm) and removal of Si precipitates, the measured remaining silica fraction is named 'dissolved silica'.

### Oxygen quantification by optode sensors

Free dissolved oxygen was quantified using a luminophore optode foil from PreSens. Small 3 by 3 mm pieces of foil were affixed with silicone glue within the lower 2 cm of the bottles to ensure direct contact with the liquid phase. The interaction between the optode foil (luminophore) and oxygen molecules (quencher) results in an energy transfer that reduces the luminescence signal of the foil. The oxygen concentration in the liquid phase was then calculated using the Stern-Volmer equation. A temperature sensor was placed in a co-incubated Schott bottle under identical experimental conditions to account for temperature variations. Each container and optode foil setup underwent a fresh two-point calibration before measurements. The calibration included one data point at full oxygen saturation and another one under oxygen-free conditions. For the first calibration point, 100 mL of pure ASW medium was added to a 250 mL Schott bottle containing the optode foil and stirred vigorously until complete $O_2$ saturation was achieved. For the second calibration point, the reductant sodium dithionite ($Na_2S_2O_3$) was added to the same bottle to eliminate all remaining dissolved oxygen.

### Intracellular ROS measurements

The membrane-permeable dye CM-H$_2$DCFDA (5-(and-6)-chloromethyl-2',7'-dichlorodihydrofluorescein diacetate, acetyl ester; Life Technologies GmbH) was used to quantify intracellular reactive oxygen species. Log-phase cultures were harvested via centrifugation and washed twice with TRIS buffer (pH 7.0). Cells were adjusted to a concentration of $5 \times 10^8$ cells/mL in TRIS buffer and degassed in the dark using a 90:10 $N_2/CO_2$ gas mixture. $FeCl_2$ was added from anoxic stock solutions to achieve final concentrations of 0, 500, 2500, and 5000 µM We then added pH-adjusted silica from a metasilicate stock solution, preheated to 90 °C in a water bath to monomerize the silica, leading to a final concentrations of 0 and 2200 µM in duplicates in two different experimental runs. The suspensions were shaken for one hour. After incubation, cells were resuspended in fresh buffer containing 5 µM CM-H$_2$DCFDA and incubated in the dark for 30 min. Following a second resuspension in fresh buffer, fluorescence emission at 519 nm (excitation at $490 \pm 5$ nm) was recorded using the infinite 200 PRO Fluorometer from Tecan Lifesciences. Background fluorescence from untreated cells and Fe minerals was subtracted to ensure accuracy.

### Cell numbers by hemocytometry

The cell density was visually counted under a Leica CTR 5500 microscope using a Neubauer-improved hemocytometer from Hirschmann. Particularly, 10 µL of sample was pipetted between sample holder and coverslip. At a magnification of 400x, the cells were counted in the smallest squares until a final count of 100 cells. 180 µL cells were fixed by the addition of 20 µL 21% PFA solution. The minerals were dissolved by the addition 200 µL of 100 mM Fe(II)-EDAS solution and 600 µL of ca. 0.8 M oxalic acid The cell count per small square was then multiplied by the dilution factor and by $2.5 \times 10^7$ as volume factor to get the real cell count in mL/L.

### MATLAB model from lab experimental data

We modeled our lab experimental data to quantify total oxygen production in the system in MATLAB (version R2023b), considering both dissolved oxygen and the contribution from the headspace. Based on the equilibrium partitioning between the aqueous solution in the head space based on the Henry-Law (see SI formula 9), oxygen production including the headspace is approximately 42 times greater than what is measurable in the dissolved liquid phase alone. Additionally, increasing cell density caused greater turbidity in the liquid phase, which led to shading effects reducing the overall photosynthetic rate per cell. We accounted for this by modeling the proportion of active versus inactive cells in the system. Detailed methods and the corresponding MATLAB code are provided in the Supplementary Information (SI).

### MATLAB model to mimic the oxygen and Fe(II) distribution in the ancient ocean

To simulate oxygen distribution in ancient ocean water columns, we used the photosynthetic rate ($r_{photo}$) derived from our lab experiments under the most realistic conditions−500 µM Fe(II) and 2200 µM silica.

**Table 3 | Photosynthetic rate (oxygen production per cell/day) of the lab experiments calculated from 6.48 × 10⁶ cells/mL ('experimental cells 1') and 1.6 × 10⁷ cells/mL ('experimental cells 2') for 10⁵ cells/mL (target cells 1) and 10⁶ cells/mL (target cells 2) in the early ocean**

| $r_{photo}$ | Experimental cells 1 | Experimental cells 2 |
|---|---|---|
| Target cells 1 | 0.289 μM/day | 2.89 μM/day |
| Target cells 2 | 0.112 μM/day | 1.12 μM/day |

We divided the experiment into two phases: Phase 1, when dissolved $Fe^{2+}$ was still present, mimicking early ocean conditions, and Phase 2, after $Fe^{2+}$ oxidation, to analyze the long-term bacterial response following $Fe^{2+}$ exposure. From Phase 1, we calculated the daily oxygen production per cell ($r_{photo}$). According to reaction (5), one molecule of $O_2$ can oxidize four $Fe^{2+}$ ions.

$$O_2 + 4Fe^{2+} + 8OH^- \leftrightarrow 4FeOOH + 2H_2O \qquad (5)$$

We determined $r_{photo}$ from the first two days of the experiment, during which no free oxygen was detected, and cyanobacteria were still in the lag phase without measurable growth, allowing the simplification that all produced oxygen was used for Fe(II) oxidation and the cell count stayed the same. Starting with 529 μM Fe(II) at day 0, 150 μM was oxidized over two days, corresponding to a rate of 75 μM/day Fe(II) oxidation, which equates to 18.75 μmol $O_2$/day based on reaction (5). Due to uncertainty in manual cell counts, we calculated $r_{photo}$ for two different cell densities observed in our experiments: $6.48 \times 10^6$ cells/mL (experimental cells 1) and $1.6 \times 10^7$ cells/mL (experimental cells 2). Reported early ocean cyanobacterial concentrations range between $10^5$cells/mL (target cells 1) and $10^6$ cells/mL (target cells 2), so we modeled four scenarios combining these experimental and target cell densities to calculate $r_{photo}$ values (see Table 3) using formula 6. Based on the study of Swanner and colleagues[12], we further estimated oxygen distribution in the early ocean water column for upwelling rates of 4, 94, and 473 m/year.

$$18.75 \, \frac{\mu M}{day} * \frac{target \; cells}{experimental \; cells} = r_{photo} \qquad (6)$$

**Reporting summary**
Further information on research design is available in the Nature Portfolio Reporting Summary linked to this article.

## Data availability
The raw data generated in this study have been deposited in the Mendeley database under https://doi.org/10.17632/wp96jybytw.2.

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

## Acknowledgements

The authors would like to thank M. Herzog for providing the data of her Bachelor Thesis. J. López Rivoldi and M. Mergenthaler are acknowledged for their experimental support. The authors would further like to thank F. Schädler and L. Grimm for maintenance and help in the laboratory. Dr. J. P. Duda, Dr. E. Runge, Dr. A. Illin, D. Gutierrez Rueda and Y. Li, are acknowledged for fruitful discussions on experimental design and data interpretation. We would like to thank the Deutsche Forschungsgemeinschaft (DFG, German Research Foundation) for supporting grants for this project to AK, project ID AOBJ: 669932.

## Author contributions

C.D., A.K. and K.K designed the experiments. M.S., A.K. and K.K. acquired the funding. C.D. performed the experiments and data analysis. O.C. provided the Matlab codes and modeled the lab data. C.D., A.K. and K.K. wrote and revised the paper, O.C. and M.S. reviewed and edited the paper.

## Funding

## Competing interests

The authors declare no competing interests.

## Additional information

**Peer review information** : *Nature Communications* thanks Kelsey Moore and the other, anonymous, reviewer(s) for their contribution to the peer review of this work. A peer review file is available.

