## [Peer Review file · Nature Communications]

Survival of cyanobacteria and mitigation of Fe(II) toxicity effects in a silica-rich Archean ocean

Corresponding Author: Professor Andreas Kappler

Version 1:

Reviewer comments:

Reviewer #1

(Remarks to the Author)

The authors investigate the hypothesis that elevated Fe(II) concentrations in the Archean oceans may have limited cyanobacterial growth, potentially explaining the ~500 million-year gap between the emergence of oxygenic photosynthesis and the Great Oxidation Event. Through a series of experiments in which a cyanobacterial strain was inoculated in media with varying Fe(II) and SiO₂ concentrations, the authors conclude that SiO₂ could have mitigated Fe toxicity in the early oceans.

This study follows up on Swanner et al. (2015a, 2015b) and adopts much of their experimental design and modeling framework. Consequently, its main novelty lies in introducing SiO₂ as an additional variable in the system. Overall, I find the study insufficiently novel. Moreover, the presentation of the results and discussion could be streamlined to improve clarity and flow, as the current structure is difficult to follow. In its present form, I cannot recommend publication in Nature Communications.

1.- While I understand the authors support of their model on oxygenic photosynthesis driving the origin of banded iron formations. The abstract and introduction include strong claims such as “Banded iron formations (BIF) were deposited... with cyanobacterial oxygen (O₂) as the main oxidant for Fe(II)(aq) oxidation” which could be nuanced to include the possibility of abiotic processes contributing to the formation of BIF deposits (e.g. photooxidation of Fe(II) or greenalite precursor models)

2.- The hypothesis regarding elevated SiO₂ conditions and their potential mitigating effects is not well motivated in the introduction, leaving the reader uncertain about why this variable is important to investigate in the first place. Much of the content in lines 456–475 does not directly discuss the results but instead introduces a relevant body of background work that should be presented earlier in the manuscript to establish a stronger rationale for the study.

3.- The authors tested multiple conditions, which increases the robustness of the study. However, the results are presented in a manner that makes it difficult for the reader to discern the major conclusions and differences between the various conditions. Since the main focus of the paper is on the effects of SiO₂ on Fe(II) toxicity, it may be more effective to first present the overall trends associated with SiO₂ presence or absence, followed by a more detailed analysis of the specific Fe(II) conditions. This approach would help the reader better appreciate the key patterns in the data.

3.- Following the same reasoning, the presentation of the figures could be improved to better facilitate comparisons between the different experimental conditions. For example, lines 145–146 state that “...This O₂ accumulation peaked earlier in the ‘no silica’ setup (after 50 days) compared to the ‘high silica’ treatment (after 60–70 days) (Fig. 1E),” yet Fig. 1E only shows the first ~10 days of data, while the complete dataset is actually presented in Fig. 2. Similar issues occur throughout the remainder of the section, making it difficult to link the authors’ claims to the data shown in the figures, as each Fe condition is plotted on a different time scale.

Similarly, the water-column modeling section presents three figures (Figures 5–7) comprising a total of 24 panels, which makes data comparison and identification of the main conclusions challenging for the reader.

Here are some suggestions that I believe would improve the figure presentation:

- Figures 1 and 2 could be combined into a single figure. It seems the authors intended to highlight the timescales at which Fe(II) concentrations approached zero in solution. I suggest using a consistent time scale for each Fe condition (e.g., displaying the full 150-day dataset) and including insets of the Fe(II) concentration curves to show the early stages in more detail (e.g., the first 10, 30, or 90 days).

- Figures 5, 6, and 7 could also be combined into a single figure, as most panels share similar Fe(II) and O₂ scales for the 10⁶–10⁷ cells/ml. Incorporating colored curves and clear legends would greatly improve the visual comparability of the data.

3.- The discussion section presents mixed results. In particular, lines 382–394 introduce a new dataset on day-night cycles well after the main results and several paragraphs of discussion, making it difficult to connect these findings to the broader implications for the system.

Minor comments/suggestions

1. Line 129.- Should be Fig S1 A, not Fig 1A

2. In Fig 3. The presented intensities relative to the abiotic and anoxic 5000 μ M Fe condition is creating negative intensities/counts for the 2500 and 5000 μ M Fe conditions, which can be confusing to the reader and at first glance might be interpreted as if there is something interfering with the spectrophotometric measurements. I would suggest normalizing the results instead to remove this artifact.

3. For the ROS measurements, why was a metasilicate solution used instead of the monomeric silica applied in the long-term experiments?

Reviewer #2

(Remarks to the Author)

This study seeks to understand the chemical factors that may have influenced the proliferation of cyanobacteria and the Great Oxidation Event, topics of broad interest to the geoscience community. The study specifically focuses on the potential connections among iron toxicity, oxidative stress, and elevated Fe(II) concentrations in Archean oceans as factors that may have limited cyanobacterial proliferation and the impact of high SiO₂ concentrations in Archean seawater as a potential mitigating factor in iron toxicity and ROS generation.

The study uses an experimental approach to test the impact of Fe(II) concentrations, oxidative stress, and SiO₂ concentrations on cyanobacterial growth and O₂ production. The authors measure O₂ concentrations, Fe(II) and SiO₂ concentrations, Fe_{total}, cell numbers, and ROS production to characterize the interactive effects of Fe(II) and SiO₂ on cyanobacterial physiology, oxygen production, and growth. These experimental findings are paired with geochemical modeling to characterize total O₂ production under Archean-type conditions and scale the laboratory results to early oceans. Together, the experimental and modeling results demonstrate how day/night cycles and elevated SiO₂ concentrations in Archean oceans could have mitigated ROS stress and Fe(II) toxicity, playing instrumental roles in the proliferation of cyanobacteria and the GOE.

Overall, the manuscript is well written, the experiments are well designed, and the results are compelling and novel. This study provides important new experimentally derived insights into the ways in which Fe(II) toxicity and mitigation of toxicity and ROS generation by SiO₂ may have impacted cyanobacterial proliferation, oxygen accumulation, and the GOE. The experimental results convincingly demonstrate that Fe(II) toxicity negatively impacts cyanobacterial growth and O₂ production. The experiments also clearly demonstrate that under elevated concentrations of SiO₂ (similar to estimates of Archean oceans), the impact of Fe(II) toxicity and ROS stress is diminished, evidenced by higher growth and O₂ production and diminished ROS in high SiO₂ cultures. Using these experimental findings, the study refines models that predict O₂ accumulation and ocean Fe(II) concentrations during the Archean and demonstrates that elevated SiO₂ concentrations in Archean oceans may have played a key role in cyanobacterial proliferation and the GOE.

The findings are novel, and the results will be of broad interest. I support publication of the manuscript, but I have several comments and suggestions outlined below and in the attached document regarding details of experimental design, presentation of the data, and potential discussion points that would strengthen the manuscript. Broadly, 1) the introduction and discussion could be strengthened by posing a possible mechanism to explain *how* silica is mitigating the impacts of Fe(II) toxicity and ROS, 2) some details of the experimental design should be mentioned in the main text (see below), 3) the data in figure 1 should be plotted on the same timescales to more robustly show trends highlighted in the text, 4) there are some mentions of the precipitates but no data characterizing what these precipitates are – if these data were collected, it would be useful to show them. These comments and other minor suggestions are outlined in the attached document with line references.

Reviewer #3

(Remarks to the Author)

Dreher et al. report new experimental data testing the effects of dissolved Fe(II) and SiO₂ on *Synechococcus* O₂ production rates. The study is a more detailed and improved version of earlier ones (e.g., Swanner et al. 2015). The study results are neat and novel and could have important implications for Archean biogeochemical cycles. The results could also shed some light on enigmatic Archean iron formation depositional conditions.

I am not an experimentalist by training; I hope at least one other reviewer is. But I do study Archean Earth. My personal

opinion is that the findings are probably novel enough to merit publication in Nature Communications. Many knowledge gaps remain unfilled in our understanding of Archean Fe cycling—and many scientists are active in this area. This paper could provide new perspective.

My most impactful comments are:

1. The authors need to dedicate at least some text in the introduction to explaining why high Archean dissolved Si concentrations are important. Why do we think this was the case, and why is that information important? This is incredibly important because it is the true novelty of these new experiments.
2. Salinity does not seem to have been considered in the O₂ solubility calculations. I think this should change.
3. How might strictly benthic Archean cyanobacteria impact their findings? Some researchers would argue that all Archean cyanos were benthic and stuck to the seafloor (i.e., not planktonic and free-floating; Sanchez-Baracaldo, 2015). How might this assumption change their hypotheses, if at all?

Some line-by-line comments:

Line 106: Why are high silica contents important for the problem? I know why, but the average reader may not. A few sentences explaining the motivation for exploring high silica seem necessary.

Line 109: Was a synthetic seawater-like solution used for the experiments? Or a freshwater solution? This information might be important enough to merit mention here, in the main text.

Line 203: This intro sentence foreshadows some discussion of ROS in the paragraph. But the ROS discussion never arrives. Is this intentional? Should the authors consider a different opening to this paragraph?

Line 282: Has anyone estimated these upwelling velocities for the Archean?

Line 287: How is salinity treated in this calculation? The solution is salinity-dependent.

Line 302: Consider stating the dissolved O₂ abundances in the surface waters for this set of calculations (you did it for the previous ones, and reading on also the subsequent ones).

Line 330: Have the authors considered calculating how much O₂ is released to the atmosphere in each scenario?

Thereafter, they may also consider thinking about whether the calculation is consistent with pO₂ estimates for the Archean (e.g., those derived from S-MIF, or detrital redox-sensitive grains).

Line 336: What are those maximum O₂ concentrations? Please state them here.

Line 343: Please cite the “earlier models”.

Line 367: An end-parentheses is missing.

Version 2:

Reviewer comments:

Reviewer #1

(Remarks to the Author)

While I agree with the other reviewers that the study is based on a robust dataset, and regardless of its potential suitability for publication in Nature Communications, I respectfully disagree with the assessment that the manuscript is well written. In its revised form, I cannot recommend the manuscript for publication. Substantial restructuring of the data presentation and the Discussion should be required before it can be considered for publication. This is not a matter of stylistic preference; rather, the study's content is not being communicated efficiently to a general readership, regardless of the quality of the underlying science. Below, I provide specific details and suggestions to improve readability.

First, I thank the authors for updating Figure 1 and for merging Figures 5, 6, and 7 into a new Figure 5. The revised figures are much easier to read, and the interpretation of the data are more readily extracted. Regarding Figure 1, I understand the authors' concerns about displaying smaller insets; however, I find that comparing datasets on identical timescales is far more effective. This approach also allows visualization of the full (long-term) Fe(II) oxidation profiles, which were not even shown in the original figure. I have several comments to further improve the readability of Figure 1. First, the updated insets appear to be downscaled versions of the original plots. As a result, the font size is extremely small and difficult to read. I recommend re-plotting these panels with increased font sizes, symbol sizes, and line thicknesses to enhance clarity. In addition, for the dissolved O₂ and cell count insets, the y-axis ranges extend well beyond what is necessary to highlight differences between conditions. For example, the original panels 1E and 1F span ranges of 1–600 and 0–3 × 10⁸, respectively, whereas adjusting these to 0–200 and 0–5 × 10⁷ would make the differences more apparent. Finally, the revised figure is missing panel labels (A through L), which should be added for clarity and ease of reference.

Despite the revisions made by the authors, I still find the main text very difficult to read. I previously suggested modifying how the results are presented in the section “Effects of different silica concentrations on Fe(II) oxidation by cyanobacterial O₂ under alternating day–night cycles.” The authors indicated that they prefer to retain the current structure. While I understand their rationale for organizing the results by Fe(II) condition and then by the presence or absence of silica, the section remains overly verbose and repetitive. As currently structured, the authors sequentially describe each Fe(II) condition, followed by color changes, cell growth, O₂ accumulation, Fe(II) oxidation, and silica precipitation, repeating this format for both silica-free and silica-amended experiments. This results in an entirely descriptive section spanning roughly ten paragraphs. Although I appreciate the authors' effort to be thorough, I feel that the key findings are obscured by the level of detail provided. In

addition, most numerical values reported often add noise rather than insight, particularly when the implications of these differences are not explicitly discussed. For example, in lines 134–138, the reported differences in final cell densities between the “no silica” and “high silica” setups are relatively small, and it is unclear whether these differences are meaningful in the context of the study. Similar examples occur throughout much of the Results section. I therefore recommend restructuring this section by (1) summarizing observations more concisely and (2) presenting key quantitative results in a table, similar to Table 1, instead of listing every single number in the main text. Such a table could report rates of cell growth and dissolved O₂ accumulation, potentially over both short and long timescales (as this timescale difference is relevant for the study). This approach would substantially improve readability while ensuring that the most important results are clearly communicated.

Along similar lines, the logic of the Discussion is very difficult to follow in its current form. For example, the authors begin by stating that “oxygenic photosynthesis on the early Earth proceeded without harmful ROS stress” before even explaining the postulated mechanisms by which ROS formation is mitigated by the presence of silica or by day–night cycles. The Discussion should be structured as a stepwise logical argument that progresses from specific experimental results to broader implications, which is not the case in its current form. For instance, it would be clearer first it is described the specific mechanisms through which silica and day–night cycling attenuate ROS formation in the authors’ experiments. This could then be followed by a broader discussion of the implications for how oxygenic photosynthesis may have proceeded on the early Earth without ROS toxicity, and how this influenced water-column redox structure and O₂ distribution in seawater.

Minor comments:

Line 3.- Would be relevant to cite “Fournier, G. P., Moore, K. R., Rangel, L. T., Payette, J. G., Momper, L., & Bosak, T. (2021). The Archean origin of oxygenic photosynthesis and extant cyanobacterial lineages. *Proceedings of the Royal Society B*, 288(1959), 20210675.” where they also support an Archean origin of total group cyanobacteria.

Line 74-75. Missing a reference.

Line 130-133. Refer to Figure 1 in this sentence.

Lines 360-364. This is the motivation and hypothesis of the study, and it was not incorporated effectively in the introduction. I would suggest exchanging these lines (360-364) to the ones in the introduction which are a summary of what is shown here: “Dreher and colleagues reported increasing Fe(II)(aq) toxicity with increasing Fe(II)(aq) concentrations, but observed markedly enhanced bacterial growth compared in the presence of silica.”

Line 380 - “Taken together, our lab coupled with the modeling results...” The sentence is missing “experiments” after “lab”.

Reviewer #2

(Remarks to the Author)

I thank the authors for their detailed responses to comments and suggestions. I think that the authors integrated all key suggestions very well and strengthened an already excellent manuscript. I have no additional comments beyond those addressed in my review and those of the other reviewers. The manuscript is well written, the results and interpretations are novel, and the findings contribute significantly to our understanding of interplays between early cyanobacteria and environmental conditions on the early Earth. I support publication of this manuscript.

Reviewer #3

(Remarks to the Author)

I am satisfied with all but one of the authors' responses to my comments. My single objection is a small one (but still important, I think). When I recommended the authors add a few lines of text stressing the importance of high Silica in Archean seawater, I was hoping to see a sentence describing the supporting data for Silica-rich Archean seawater. Archean seawater WAS silica rich; this was the point I wanted them to make clearer. My apologies for the unclear directions.

Response to Reviews

Response to Reviewer #1:

(Remarks to the Author): The authors investigate the hypothesis that elevated Fe(II) concentrations in the Archaean oceans may have limited cyanobacterial growth, potentially explaining the ~500 million-year gap between the emergence of oxygenic photosynthesis and the Great Oxidation Event. Through a series of experiments in which a cyanobacterial strain was inoculated in media with varying Fe(II) and SiO₂ concentrations, the authors conclude that SiO₂ could have mitigated Fe toxicity in the early oceans. This study follows up on Swanner et al. (2015a, 2015b) and adopts much of their experimental design and modeling framework. Consequently, its main novelty lies in introducing SiO₂ as an additional variable in the system. Overall, I find the study insufficiently novel. Moreover, the presentation of the results and discussion could be streamlined to improve clarity and flow, as the current structure is difficult to follow. In its present form, I cannot recommend publication in Nature Communications.

We thank reviewer 1 for the helpful feedback and have revised our manuscript accordingly, see detailed comments below.

1. While I understand the authors support of their model on oxygenic photosynthesis driving the origin of banded iron formations. The abstract and introduction include strong claims such as “Banded iron formations (BIF) were deposited... with cyanobacterial oxygen (O₂) as the main oxidant for Fe(II)(aq) oxidation” which could be nuanced to include the possibility of abiotic processes contributing to the formation of BIF deposits (e.g. photooxidation of Fe(II) or greenalite precursor models)

We changed the abstract and introduction following the reviewer’s recommendation to make sure other deposition pathways are sufficiently credited.

Lines 24-26: “Banded iron formations (BIF) were deposited abundantly between 2.7-2.4 Ga from iron- and silica-rich oceans, with cyanobacterial oxygen (O₂) as a possible oxidant for Fe(II)_(aq) oxidation and Fe(III) oxyhydroxide precipitation.”

Lines 55-58: “While the deposition of the oldest BIF was probably linked to anoxygenic phototrophic Fe(II)-oxidizers^{22,23}, photochemical Fe(II) oxidation^{24,25} or abiotic deposition processes such as chemical greenalite deposition^{26,27}, the proliferation of cyanobacteria would have accelerated Fe(II) oxidation and thus BIF deposition^{12,28}.”

2. The hypothesis regarding elevated SiO_2 conditions and their potential mitigating effects is not well motivated in the introduction, leaving the reader uncertain about why this variable is important to investigate in the first place. Much of the content in lines 456–475 does not directly discuss the results but instead introduces a relevant body of background work that should be presented earlier in the manuscript to establish a stronger rationale for the study.

The authors agree that the motivation of using SiO_2 should be mentioned earlier in the introduction and its role in mitigating potential Fe(II)-related toxicity emphasized. We, therefore, revised the introduction as follows:

Lines 108-110: “Dreher and colleagues³⁴ reported increasing $\text{Fe(II)}_{(\text{aq})}$ toxicity with increasing $\text{Fe(II)}_{(\text{aq})}$ concentrations, but observed markedly enhanced bacterial growth compared in the presence of silica. Based on these results, we hypothesize that silica reacts with $\text{Fe(II)}_{(\text{aq})}$, thus distinctly lowering Fe(II) reactivity and preventing Fe(II)-induced ROS formation. To test this hypothesis, we experimentally examined the combined effects of $\text{Fe(II)}_{(\text{aq})}$ and $\text{SiO}_{2(\text{aq})}$ on cyanobacterial O_2 production, Fe(II) oxidation, and potential ROS-related toxicity.”

3.-The authors tested multiple conditions, which increases the robustness of the study. However, the results are presented in a manner that makes it difficult for the reader to discern the major conclusions and differences between the various conditions. Since the main focus of the paper is on the effects of SiO_2 on Fe(II) toxicity, it may be more effective to first present the overall trends associated with SiO_2 presence or absence, followed by a more detailed analysis of the specific Fe(II) conditions. This approach would help the reader better appreciate the key patterns in the data.

We appreciate the reviewer’s suggestion to restructure the Results section and note that we explored this approach by beginning directly away with the effect of Si on Fe(II) toxicity. However, this organization immediately raises a fundamental question for the reader: “what is the baseline level of O_2 production under “healthy” conditions, i.e., in the absence of both Fe(II) and Si, against which all other treatments are compared?”. For this reason, we believe it is clearer to retain the current structure and to present the results systematically, starting with the unamended control experiments (no Fe(II), no Si), followed by experiments with increasing Fe(II) concentrations in the absence of Si, and finally those with Fe(II) in the presence of Si. This progression allows the rescuing effect of Si on Fe(II) toxicity to be demonstrated unambiguously. We hope that this rationale is acceptable to the editor and reviewer.

3.- Following the same reasoning, the presentation of the figures could be improved to better facilitate comparisons between the different experimental conditions. For example, lines 145–146 state that “...This O_2 accumulation peaked earlier in the ‘no silica’ setup (after 50 days) compared to the ‘high silica’ treatment (after 60–70 days) (Fig. 1E),” yet Fig. 1E only shows the first ~10 days of data, while the complete dataset is actually presented in Fig. 2. Similar issues occur throughout the remainder of the section, making it difficult to link the authors’ claims to the data shown in the figures, as each Fe condition is plotted on a different time scale.

Similarly, the water-column modeling section presents three figures (Figures 5–7) comprising a total of 24 panels, which makes data comparison and identification of the main conclusions challenging for the reader.

We agree with the reviewer that plotting datasets on different timescales may initially appear counterintuitive. However, if all experiments were displayed on a uniform 90-day x-axis (as in the 5000 μM setup), the differences in Fe(II) oxidation rates would be obscured. The timescale most relevant for the reader is the interval of active Fe(II) oxidation by the cyanobacteria, and we therefore focus on (i.e., zoom into) this period. To avoid confusion, we have clarified this rationale in the figure caption and, following the reviewer’s suggestion, explicitly refer to Fig. 2, where the longer-term effects on cell counts and oxygen production are presented.

Lines 145-150: “Fig. 1: Geochemical data of Fe(II) oxidation experiments containing *Synechococcus* sp. PCC 7002. Panels A to C show the results of setups without amendment of Fe(II)(aq). Panels D to F show the results of setups with amendment of 500 μM Fe(II)(aq), panels G to I with 2500 μM Fe(II)(aq), and panels J to L with 5000 μM Fe(II)(aq). Filled symbols indicate setups with 2200 μM silica (‘high silica’), empty symbols indicate setups without silica (‘no silica’). Fe(II) (grey), accumulated oxygen (blue), cell numbers (green) are plotted as average values in three triplicates with the standard deviation as error bars of the experiments. Please note that the time scale (x-axis) varies for the different plots because of the differences in the duration of active Fe(II) oxidation.”

Lines 132-135: “The experiments were plotted in the respective timescale of the active Fe(II) oxidation, as this sets the focus on the respective period with Fe(II)(aq) in the system, which is the most relevant for determining toxicity effects and the latter modelling.”

Here are some suggestions that I believe would improve the figure presentation:

- Figures 1 and 2 could be combined into a single figure. It seems the authors intended to highlight the timescales at which Fe(II) concentrations approached zero in solution. I suggest using a consistent time scale for each Fe condition (e.g., displaying the full 150-day dataset) and including insets of the Fe(II) concentration curves to show the early stages in more detail (e.g., the first 10, 30, or 90 days).
- Figures 5, 6, and 7 could also be combined into a single figure, as most panels share similar Fe(II) and O₂ scales for the 10⁶–10⁷ cells/ml. Incorporating colored curves and clear legends would greatly improve the visual comparability of the data.

We understand the reviewer’s point and generally agree that using identical timescales across plots is, in principle, desirable. Accordingly, we prepared the figure following the reviewer’s suggestion (see figure provided below this response). However, our assessment is that the revised version is less effective, because when all experiments are shown on the same timescale the key dynamics occurring over very short intervals (particularly in the 500 and 2500 μM Fe(II) treatments) become visually indistinguishable. As a result, the main panels occupy substantial space while conveying little information, and the reader is effectively forced to rely on the inset plots to see where the relevant processes occur. For this reason, we believe it is clearer to retain Figures 1 and 2 in their original form, while more explicitly explaining the rationale for the different timescales in the captions and improving cross-referencing to specific figures and panels in the main text. We note that Figures 1 and 2 were designed to address distinct aspects of the study: Figure 1 focuses on the period of active Fe(II) oxidation, whereas Figure 2 illustrates the longer-term toxicity effects. Nevertheless, if the editor prefers the revised figure with uniform timescales, we are of course happy to adopt that format. Finally, following the reviewer’s suggestion, we have combined figures 5, 6 and 7.

3.- The discussion section presents mixed results. In particular, lines 382–394 introduce a new dataset on day-night cycles well after the main results and several paragraphs of discussion, making it difficult to connect these findings to the broader implications for the system.

We would like to thank the reviewer for this comment and are happy to clarify this point. The dataset referred by the reviewer actually stems from a separate study. In the meantime, since submission of the present paper to Nature Communications, the paper containing the mentioned dataset has been published and is now cited in the text (Dreher et al., 2025).

Lines 389-391: “In these continuous light experiments, the presence of 2200 μM silica resulted in remarkably higher cell counts and oxygen concentrations compared to the silica-free equivalents³⁴.”

Minor comments/suggestions

1. Line 129.- Should be Fig S1 A, not Fig 1A

Changed, thank you for noticing.

2. In Fig 3. The presented intensities relative to the abiotic and anoxic 5000 uM Fe condition is creating negative intensities/counts for the 2500 and 5000 uM Fe conditions, which can be confusing to the reader and at first glance might be interpreted as if there is something interfering with the

spectrophotometric measurements. I would suggest normalizing the results instead to remove this artifact.

We would like to thank the reviewer for this comment. As noted, all data presented are already normalized: specifically, all counts were normalized to an abiotic, anoxic reference setup containing 5000 μM Fe(II) and 2200 μM of SiO_2 . Consequently, the negative values identified by the reviewer indicate that counts measured in the biotic setup (i.e., those including cyanobacterial cells) were lower than those in the abiotic setup. Since the data are normalized, it is important to show the negative values. However, we agree that this might initially be confusing for the reader, and we have therefore explained this in more detail in the figure caption, as follows:

Lines 247-249: “and presence of 2200 μM $\text{SiO}_{2(\text{aq})}$ (orange). Please note that the normalization of all measurements to the abiotic control can result in negative values. Error bars show the standard....”.

3. For the ROS measurements, why was a metasilicate solution used instead of the monomeric silica applied in the long-term experiments?

The original mixture consisted of a metasilicate solution in MQ water, which was then heated in a water bath to 90°C to create monomeric silica. We now clarified this in the method section.

Lines 572-575: “ FeCl_2 was added from anoxic stock solutions to achieve final concentrations of 0, 500, 2500 and 5000 μM . We then added pH-adjusted silica from a metasilicate stock solution preheated to 90°C in a water bath to monomerize the silica, leading to a final concentrations of 0 and 2200 μM in duplicates in two different experimental runs.”

Response to Reviewer #2

(Remarks to the Author): This study seeks to understand the chemical factors that may have influenced the proliferation of cyanobacteria and the Great Oxidation Event, topics of broad interest to the geoscience community. The study specifically focuses on the potential connections among iron toxicity, oxidative stress, and elevated Fe(II) concentrations in Archean oceans as factors that may have limited cyanobacterial proliferation and the impact of high SiO₂ concentrations in Archean seawater as a potential mitigating factor in iron toxicity and ROS generation.

The study uses an experimental approach to test the impact of Fe(II) concentrations, oxidative stress, and SiO₂ concentrations on cyanobacterial growth and O₂ production. The authors measure O₂ concentrations, Fe(II) and SiO₂ concentrations, Fetotal, cell numbers, and ROS production to characterize the interactive effects of Fe(II) and SiO₂ on cyanobacterial physiology, oxygen production, and growth. These experimental findings are paired with geochemical modeling to characterize total O₂ production under Archean-type conditions and scale the laboratory results to early oceans. Together, the experimental and modeling results demonstrate how day/night cycles and elevated SiO₂ concentrations in Archean oceans could have mitigated ROS stress and Fe(II) toxicity, playing instrumental roles in the proliferation of cyanobacteria and the GOE.

Overall, the manuscript is well written, the experiments are well designed, and the results are compelling and novel. This study provides important new experimentally derived insights into the ways in which Fe(II) toxicity and mitigation of toxicity and ROS generation by SiO₂ may have impacted cyanobacterial proliferation, oxygen accumulation, and the GOE. The experimental results convincingly demonstrate that Fe(II) toxicity negatively impacts cyanobacterial growth and O₂ production. The experiments also clearly demonstrate that under elevated concentrations of SiO₂ (similar to estimates of Archean oceans), the impact of Fe(II) toxicity and ROS stress is diminished, evidenced by higher growth and O₂ production and diminished ROS in high SiO₂ cultures. Using these experimental findings, the study refines models that predict O₂ accumulation and ocean Fe(II) concentrations during the Archean and demonstrates that elevated SiO₂ concentrations in Archean oceans may have played a key role in cyanobacterial proliferation and the GOE.

The findings are novel, and the results will be of broad interest. I support publication of the manuscript, but I have several comments and suggestions outlined below and in the attached document regarding details of experimental design, presentation of the data, and potential discussion points that would strengthen the manuscript. Broadly, 1) the introduction and discussion could be strengthened by posing a possible mechanism to explain *how* silica is mitigating the impacts of Fe(II) toxicity and ROS, 2) some details of the experimental design should be mentioned in the main text (see below), 3) the data in figure 1 should be plotted on the same timescales to more robustly show trends highlighted in the text, 4) there are some mentions of the precipitates but no data characterizing what these precipitates are – if these data were collected, it would be useful to show them. These comments and other minor suggestions are outlined in the attached document with line references.

We thank reviewer #2 for the positive and detailed feedback. Please find below our responses to all comments.

1. Line 106: The introduction does a great job of painting a picture both of the connection between Fe(II) and oxidative stress, the potential mitigation of these impacts by elevated SiO₂ concentrations, and relationship between Fe and SiO₂ in BIFs indicating iron-silica interactions in Archean oceans. However, the specific connection between Fe(II) stress and SiO₂ mitigation and the potential underlying mechanism is not clear. What is the specific hypothesis or hypothesized mechanism being tested? That Si may remove FeII? Or that it may directly interact with the cells in some way? Mentioning the hypothesized mechanism in the introduction and the discussion (see later comments)

would be of interest to the community and strengthen the findings from the study.

The authors agree with this comment. Following the reviewer's suggestion, we now explicitly state the hypothesis that silica and Fe(II)_{aq} interact with each other, therefore partly preventing or at least distinctly slowing down the Fenton's reaction of Fe(II)_{aq} with O₂.

Line 108-114: "Dreher and colleagues³⁴ reported increasing Fe(II)_(aq) toxicity with increasing Fe(II)_(aq) concentrations, but observed markedly enhanced bacterial growth compared in the presence of silica. Based on these results, we hypothesize that silica reacts with Fe(II)_(aq), thus distinctly lowering Fe(II) reactivity and preventing Fe(II)-induced ROS formation. To test this hypothesis, we experimentally examined the combined effects of Fe(II)_(aq) and SiO_{2(aq)} on cyanobacterial O₂ production, Fe(II) oxidation, and potential ROS-related toxicity."

2. Line 120: Although the experimental design is outlined in the results section, it would be useful to include a brief note at the beginning of the results section to help the reader follow the results. Just a sentence or two noting the basic setup, number of replicates, and variables being tested.

We agree that this would be useful and therefore added two sentences in the beginning of the results section. Additionally, we added the information that the experiments were performed in triplicates to the figure captions of Fig. 1 and 2.

Lines 126-129: "We conducted Fe(II) oxidation experiments, under initially anoxic conditions, using Fe(II)_(aq) concentrations ranging from 0.5 to 5 mM. Following inoculation with the cyanobacterial strain *Synechococcus* PCC 7002, we monitored Fe(II), Fe(tot), cyanobacterially produced O₂, dissolved Si, and cell counts over time (for sterile controls see Fig. S3)."

Lines 147-149 +230-241-232: "Filled symbols indicate setups with 2200 μM silica ('high silica'), while empty symbols indicate setups without silica ('no silica'). Fe(II) (grey), accumulated oxygen (blue), cell numbers (green) are plotted as average values from triplicates with the standard deviations shown as error bars."

3. Results/Methods: Related to this, it is not clear between the initial growth conditions and experimental conditions whether the experiments were conducted under anoxic conditions. It would be helpful to explicitly state this at the beginning of the results section as this is a key parameter. Whether the experiments were conducted under oxic or anoxic conditions has strong implications for Fe oxidation both at time 0 and over the course of the experiments as well as the changes in O₂ observed.

The authors agree that it is important to point out that all oxygen measured during the experiment is the result of cyanobacterial activity. We now explicitly mention this in text.

Lines 126-129: "We conducted Fe(II) oxidation experiments, under initially anoxic conditions, using Fe(II)_(aq) concentrations ranging from 0.5 to 5 mM. Following inoculation with the cyanobacterial strain *Synechococcus* PCC 7002, we monitored Fe(II), Fe(tot), cyanobacterially produced O₂, dissolved Si, and cell counts over time (for sterile controls see Fig. S3)."

Lines 524-526: "The experiments were initially fully anoxic, so all oxygen detected during the course of the experiments was produced solely by the inoculated cyanobacteria."

4. Results/Methods: Also related to the experimental design, were sterile controls used to confirm that Fe(II) was not oxidized in the absence of cyanobacteria?

Thanks for this excellent question. We added plots of the sterile (abiotic) control setups to the SI. We ran two anoxic controls in parallel, one containing 5000 μM Fe(II)_(aq) and 2200 μM silica, the other one containing no additional Fe(II)_(aq) and no silica.

Fig.S3: "Sterile controls of the Fe(II) oxidation experiments. Panel A shows the results of setups without amendment of Fe(II)_(aq) or Si. Panel B shows the results of setups with amendment of 5000 μM Fe(II)_(aq) and 2200 μM Si."

Lines 127-129. "Following inoculation with the cyanobacterial strain *Synechococcus* sp. PCC 7002, we monitored Fe(II), Fe(total), cyanobacterially produced O₂, dissolved silica, and cell counts over time (for sterile controls see Fig. S3)."

5. Line 126 and Figure 1A: Why does Fe(II) increase in the control condition by the end of the experiment?

Thank you for this comment. The total Fe(II) indeed increased from ca. 40 μM to approximately 60 μM . Since in these experiments no Fe(II) was added like in the 500 μM , 2500 μM and 5000 μM Fe(II) experiments, the detected Fe(II) exclusively stems from the trace metal solution of the growth medium. We assume the fluctuation in detected concentrations is related to precipitation (and remobilization) of Fe(II) with phosphate or silica at the glass wall. We have detected this before (Notini et al., 2019).

6. Figure 1: Why are the plots in Fig 1 plotted on different timescales? It would be more useful to directly compare the experiments over the same timescales (x axis) and then re-scale the Y axes so that small changes (like those in A, E, F, and I) can be more clearly seen.

See also response to a similar comment by reviewer #1: We agree that it might seem a bit counterintuitive to plot all the datasets on different timescales. However, if every setup would be plotted on an x-axis timescale of 90 days (like the 5000 μM setup), the differences in the Fe(II) oxidation rates would disappear completely. The timescale that is relevant for the reader, however, is the timeframe of active Fe(II) oxidation by the cyanobacteria - thus is why we zoomed in into this period. To clarify this and avoid confusion, we have now clarified this in the figure caption and also directly refer to "Fig. 2", where the long-term effect on the cell counts and oxygen production are described.

Lines 145-150: "Fig. 1: Geochemical data of Fe(II) oxidation experiments containing *Synechococcus* sp. PCC 7002. Panels A to C show the results of setups without amendment of Fe(II)_(aq). Panels D to F show the results of setups with amendment of 500 μM Fe(II)_(aq), panels G to I with 2500 μM Fe(II)_(aq), and panels J to L with 5000 μM Fe(II)_(aq). Filled symbols indicate setups with 2200 μM silica ('high silica'), empty symbols indicate setups without silica ('no silica'). Fe(II) (grey), accumulated oxygen (blue), cell numbers (green) are plotted as average values in three triplicates with the standard deviation as error bars of the experiments. Please note that the time scale (x-axis) varies for the different plots because of the differences in the duration of active Fe(II) oxidation."

7. Line 148: Although error is stated in the table, it would be helpful to state in the text to make it clear that the oxidation rates at this concentration of silica and Fe(II) were similar within error. As written, it seems to imply that there is a difference in the oxidation rates with and without silica, but these values are similar within error at the 500 μM Fe(II) concentration based on the table.

Agreed, we added this information.

Lines 159-160: "With and without silica, the Fe(II) oxidation rates were similar within the calculated error (Fehler! Verweisquelle konnte nicht gefunden werden)."

8. Line 163: The color changes do not seem to be the key result, especially given the potential confounding variables and mechanisms underpinning the color changes that the authors note. These "color change" results could be moved to the supplemental document so that the results of the main text could more clearly focus on the key quantifiable trends.

We would like to thank the reviewer for this feedback and understand that the color changes are not a quantitative variable. However, in the incubations with cyanobacteria and Fe(II)/Si, the color changes are very obvious and clearly indicate cyanobacterial growth and activity (O_2 production leading to Fe(II) oxidation). Because the main goal of the study was to demonstrate how Si rescues the Fe(II)-influenced activity of the cyanobacteria, we consider the color changes as an easy but very illustrative way of following the cyanobacterial activity. We would therefore prefer to keep this information in the text, as it might be very useful for other scientists who would like to do similar work.

9. Line 182: Is Fe still getting oxidized in the high silica experiments? If so, why were Fe(III) precipitates (i.e., orange colors) not observed? Or is the Fe(II) being complexed by the silica? Do you have any data (e.g., SEM/EDS) characterizing the nature of the precipitates? This could be helpful in

understanding the underlying mechanism by which silica may be buffering against Fe(II) toxicity.

Yes, the Fe(II) was still getting oxidized in the high Si experiments (the observed green color just indicates that the cyanobacteria grew very well even in the presence of high Fe(II)_(aq). This is described in the following lines where we wrote:

Lines 207-211: “The Fe(II) oxidation rates were similar between the ‘no silica’ and ‘high silica’ setups—77 and 72 $\mu\text{M}/\text{day}$, respectively (**Fehler! Verweisquelle konnte nicht gefunden werden. J; Fehler! Verweisquelle konnte nicht gefunden werden.**) —but notably lower than the rates observed in experiments with 2500 μM Fe(II)_(aq). In the ‘high silica’ setups, the initial SiO_{2(aq)} concentration of 1500 μM declined to 500 μM by day 31 (**Fehler! Verweisquelle konnte nicht gefunden werden. D**), after which it either stabilized or continued to precipitate, reaching final concentrations between 100 and 200 μM . “

In the present study, we unfortunately did not use SEM/EDS for analysis of the precipitates. However, the mineral composition should be comparable to what we found in our previous study (Dreher et al. 2025, GCA), where we found evidence for the co-precipitation of Si and Fe in aggregates. We now cite this study in the text.

Lines 360-364: “Our working hypothesis is that silica binds Fe(II)_{aq}, forming Fe-Si-aggregates that slow Fe(II) oxidation and consequently inhibit ROS formation. Previous studies have shown that slower Fe(II) oxidation rates reduce ROS formation³⁸. Supporting this, Dreher and colleagues³⁴ showed, using SEM/EDS, that Fe-Si aggregates precipitate under similar experimental conditions, further strengthening our hypothesis.”

10. Line 189: Do you have a hypothesis to explain this variability?

We thank the reviewer for this comment. We attribute the observed variability in O₂ accumulation between different bottles to minor differences in setup composition, such as small variations in inoculum cell numbers or in glass surface properties affecting trace metal sorption and precipitation. Such variability is common in microbial experiments that include minerals and metals. However, we believe adding that discussing this in detail would detract from the main focus of our study, so we have chosen not to include it in our manuscript.

11. Line 231: Why was a different starting concentration of silica used in these experiments compared to the growth experiments (1600 μM compared to 2200 μM)?

Thank you for pointing this out. We understand that this is confusing. However, in both experiments 2200 μM Si was added. The missing fraction of the silica was in the solid fraction. We filtered the samples; therefore, we name the measured Si fraction “dissolved silica”. This is now explained in the method section.

Lines 549-551: “Due to the filtration of the samples (0.22 μm) and removal of Si precipitates, the measured remaining silica fraction is named ‘dissolved silica’.”

12. Line 363: Here, as in the introduction, you could consider posing a potential mechanism to explain *how* silica mitigates the impact of Fe(II) toxicity and ROS. The results of the study are very compelling and demonstrate that there is a correlation and that SiO₂ may have been a key component to Fe (II) toxicity and ROS mitigation, but these results would be even more impactful if you outlined some potential mechanisms.

We agree with the reviewer and have therefore added a paragraph to explain the underlying hypothesis better.

Lines 360-364: “Our working hypothesis is that silica binds Fe(II)_{aq}, forming Fe-Si-aggregates that slow Fe(II) oxidation and consequently inhibit ROS formation. Previous studies have shown that slower Fe(II) oxidation rates reduce ROS formation³⁶. Supporting this, Dreher and colleagues³² showed, using SEM/EDS, that Fe-Si aggregates precipitate under similar experimental conditions, further strengthening our hypothesis.”

13. Line 414: Do you think that the length of the day matters?

This is an excellent question. As we did not vary different day lengths in our experiments, we cannot draw any definitive conclusion regarding this question. However, other research suggests that there most likely is a day-night-optimum for cyanobacteria (e.g. Muhetaer et al.2020). We discuss research regarding the influence of day-night-cycles on photoferrotrophic bacteria and cyanobacteria in lines 403-427.

14. Line 458: Do you have any data aside from observations of cloudiness in the bottles to characterize the precipitates?

We did not analyze the precipitates in this study. However, in our previous study (Dreher et al. 2025, GCA), we analyzed the precipitates during iron cycling experiments in detail and found the precipitates to contain iron and silica. We therefore added this important information and cite Dreher et al. (2025).

Lines 464-469: “The protection effect of $\text{SiO}_{2(\text{aq})}$ against ROS formation can be explained by the ability of Si to bind $\text{Fe(II)}_{(\text{aq})}$, yielding Fe(II)-Si complexes^{27,34} that lower Fe(II) reactivity (with O_2) and thus inhibit/reduce Fenton-type reactions and ROS production. The formation of amorphous Fe(II)-Si aggregates hypothesized in our experiments correlates well with previous studies simulating Archean ocean conditions (1 mM $\text{SiO}_{2(\text{aq})}$, 1.1 mM $\text{Fe(II)}_{(\text{aq})}$, pH of 6.5–7.5; ^{60,61}).”

Response to Reviewer #3

(Remarks to the Author): Dreher et al. report new experimental data testing the effects of dissolved Fe(II) and SiO₂ on Synechococcus O₂ production rates. The study is a more detailed and improved version of earlier ones (e.g., Swanner et al. 2015). The study results are neat and novel and could have important implications for Archean biogeochemical cycles. The results could also shed some light on enigmatic Archean iron formation depositional conditions.

I am not an experimentalist by training; I hope at least one other reviewer is. But I do study Archean Earth. My personal opinion is that the findings are probably novel enough to merit publication in Nature Communications. Many knowledge gaps remain unfilled in our understanding of Archean Fe cycling—and many scientists are active in this area. This paper could provide new perspective.

The authors thank reviewer #3 for the positive feedback.

My most impactful comments are:

1. The authors need to dedicate at least some text in the introduction to explaining why high Archean dissolved Si concentrations are important. Why do we think this was the case, and why is that information important? This is incredibly important because it is the true novelty of these new experiments.

We agree and have therefore added the requested information to the introduction.

Lines 107-114: “However, the potential mitigating impact of high SiO_{2(aq)} concentrations on Fe(II)_(aq) toxicity has not been previously considered. Dreher and colleagues³⁴ reported increasing Fe(II)_(aq) toxicity with increasing Fe(II)_(aq) concentrations, but observed markedly enhanced bacterial growth compared in the presence of silica. Based on these results, we hypothesize that silica reacts with Fe(II)_(aq), thus distinctly lowering Fe(II) reactivity and preventing Fe(II)-induced ROS formation. To test this hypothesis, we experimentally examined the combined effects of Fe(II)_(aq) and SiO_{2(aq)} on cyanobacterial O₂ production, Fe(II) oxidation, and potential ROS-related toxicity.”

2. Salinity does not seem to have been considered in the O₂ solubility calculations. I think this should change.

What is typically meant by solubility of oxygen in water is the aqueous-phase concentration of oxygen in equilibrium with the atmosphere. This equilibrium is affected by salinity (sometimes referred to as “salting out”). However, our model assumes that the atmospheric oxygen concentration is zero, so that this equilibrium is irrelevant. There could be an effect at very high oxygen concentrations: If the sum of partial pressures of all volatile compounds exceeds the fluid pressure (which increases linearly with depth), the water is supersaturated with respect to dissolved gases, and bubbles can form.

These bubbles rise. With a continuous production of oxygen, this would lead to a layer with oxygen concentrations that are in local equilibrium with the bubbles. Excess oxygen ultimately vents to the atmosphere. We have not included this mechanism in our model.

Lines 300-304: “Fig. 5: Modelled oxygen (left, blue) and Fe(II) (right, grey) concentrations in the ancient ocean water column for an upwelling velocity of 4 (panels A-D), 95 (panels E-H) and 473 m/yr (panels I-L). Panels A, C, E, G, I and K are calculated for hypothesized 10⁵ cells/mL in the early ocean, B, D, F, H, J, L for 10⁶ cells/mL. Panels A, B, E, F, I, and J are modeled based on the photosynthetic rate (r_{photo}) calculated from the lowest cell count we measured in our experiments at d0; C, C, G, H, K and L are based on the highest cell count we measured at d0.”

3. How might strictly benthic Archean cyanobacteria impact their findings? Some researchers would argue that all Archean cyanos were benthic and stuck to the seafloor (i.e., not planktonic and free-floating; Sanchez-Baracaldo, 2015). How might this assumption change their hypotheses, if at all?

In our lab experiments, it likely makes little difference whether benthic or planktonic cyanobacteria were used, as the small bottle volumes allow oxygen to rapidly diffuse throughout the liquid and headspace. By contrast, the 1-D steady-state model of the ocean cannot handle benthic organisms as

it does not reach down to the seafloor. In this model the photic zone does not reach the seafloor (no sunlight present at the seafloor). As a consequence, restricting Archean photosynthesis to benthos implies restricting it to very shallow shelf regions.

Some line-by-line comments:

4. Line 106: Why are high silica contents important for the problem? I know why, but the average reader may not. A few sentences explaining the motivation for exploring high silica seem necessary.

The authors agree that pointing out the importance of the silica in the system is necessary.

Lines 107-114: "However, the potential mitigating impact of high $\text{SiO}_{2(\text{aq})}$ concentrations on $\text{Fe(II)}_{(\text{aq})}$ toxicity has not been previously considered. Dreher and colleagues³⁴ reported increasing $\text{Fe(II)}_{(\text{aq})}$ toxicity with increasing $\text{Fe(II)}_{(\text{aq})}$ concentrations, but observed markedly enhanced bacterial growth compared in the presence of silica. Based on these results, we hypothesize that silica reacts with $\text{Fe(II)}_{(\text{aq})}$, thus distinctly lowering Fe(II) reactivity and preventing Fe(II) -induced ROS formation. To test this hypothesis,, we experimentally examined the combined effects of $\text{Fe(II)}_{(\text{aq})}$ and $\text{SiO}_{2(\text{aq})}$ on cyanobacterial O_2 production, Fe(II) oxidation, and potential ROS-related toxicity"

5. Line 109: Was a synthetic seawater-like solution used for the experiments? Or a freshwater solution? This information might be important enough to merit mention here, in the main text.

We conducted the experiments in artificial seawater medium and added the information accordingly.

Lines 114-118: "Specifically, we incubated *Synechococcus* sp. PCC 7002 under alternating day-night-cycles (16 h light; 8 h dark) in artificial seawater medium, in the absence of $\text{SiO}_{2(\text{aq})}$ (0 μM , which we refer to as 'no-silica') and in the presence of $\text{SiO}_{2(\text{aq})}$ (2200 μM , which we refer to as 'high-silica') and varying initial $\text{Fe(II)}_{(\text{aq})}$ concentrations (0, 500, 2500, 5000 μM)."

6. Line 203: This intro sentence foreshadows some discussion of ROS in the paragraph. But the ROS discussion never arrives. Is this intentional? Should the authors consider a different opening to this paragraph?

It was not intentional to foreshadow into a discussion at this point of the manuscript, as results and discussion are strictly separated. We changed the sentence.

Lines 213-216: "In addition to experiments focusing on cyanobacterial growth and activity (i.e., cell numbers, O_2 production, resulting Fe(II) oxidation and dissolved silica (see Fig. S1), we explored the role of silica in promoting long-term cell viability experiments by observing potentially mitigation effects of the harmful effects of ROS generated in the presence of both $\text{Fe(II)}_{(\text{aq})}$ and O_2 ."

7. Line 282: Has anyone estimated these upwelling velocities for the Archean?

The authors are not aware of trustworthy calculations, that is why variables of modern oceans (like the black sea, or the Atlantic ocean) were used (See Table S4).

8. Line 287: How is salinity treated in this calculation? The solution is salinity-dependent.

See answer to question 2 (by the same reviewer).

9. Line 302: Consider stating the dissolved O_2 abundances in the surface waters for this set of calculations (you did it for the previous ones, and reading on also the subsequent ones).

We agree and changed the text accordingly to include the respective oxygen concentrations.

Lines 306-313: "By increasing the upwelling rate to 95 m yr^{-1} , our model predicts that, at a cyanobacterial abundance of 10^5 cells mL^{-1} (Fig. 6 A, C), $\text{Fe(II)}_{(\text{aq})}$ would have reached the ocean surface. Oxygen concentrations at the surface would have reached between 0.04 to 0.4 μM maximum concentration. At a higher theoretical abundance of 10^6 cells mL^{-1} (Fig. 6 B, D), however, the photic zone remained $\text{Fe(II)}_{(\text{aq})}$ -free down to depths of ~ 200 m or ~ 125 m, respectively. In these high cell density scenarios, O_2 accumulated to much higher concentrations within the upper 0–50 m of the water column, with modelled values reaching (200 μM) or exceeding (800 μM) maximum O_2 saturation."

10. Line 330: Have the authors considered calculating how much O₂ is released to the atmosphere in each scenario? Thereafter, they may also consider thinking about whether the calculation is consistent with pO₂ estimates for the Archean (e.g., those derived from S-MIF, or detrital redox-sensitive grains).

The oxygen release to the atmosphere is a direct outcome of the model (effective diffusion coefficient times the vertical concentration gradient at the ocean surface. As stated above (and in the manuscript) the model assumes that the atmosphere is free of oxygen. Evaluating the evolution of pO₂ would require coupling our model to an atmosphere model that includes terrestrial oxygen sinks. This is beyond the scope of our study.

11. Line 336: What are those maximum O₂ concentrations? Please state them here.

We added the requested information.

Lines 333-337: “Incorporating the upwelling rates and Fe(II)_(aq) concentrations applied in the studies of Swanner and colleagues¹¹ into our models (see Fig. S4-S6), and applying our most conservative O₂ production rate (r_{photo} of 0.122 μM/day), yields maximum O₂ concentrations at least an order of magnitude greater than previously estimated (33 μM of maximum oxygen compared to 2-4 μM of Swanner and colleagues¹¹)”

12. Line 343: Please cite the “earlier models”.

Changed.

Line 342: Citation: “Swanner and colleagues (2015): Modulation of oxygen production in Archaean oceans by episodes of Fe(II) toxicity”.

13. Line 367: An end-parentheses is missing.

Corrected.

Response to Reviews

Response to Reviewer #1:

(Remarks to the Author): While I agree with the other reviewers that the study is based on a robust dataset, and regardless of its potential suitability for publication in Nature Communications, I respectfully disagree with the assessment that the manuscript is well written. In its revised form, I cannot recommend the manuscript for publication. Substantial restructuring of the data presentation and the Discussion should be required before it can be considered for publication. This is not a matter of stylistic preference; rather, the study's content is not being communicated efficiently to a general readership, regardless of the quality of the underlying science. Below, I provide specific details and suggestions to improve readability.

We thank reviewer #1 for the constructive feedback and the effort to revise our manuscript.

First, I thank the authors for updating Figure 1 and for merging Figures 5, 6, and 7 into a new Figure 5. The revised figures are much easier to read, and the interpretation of the data are more readily extracted. Regarding Figure 1, I understand the authors' concerns about displaying smaller insets; however, I find that comparing datasets on identical timescales is far more effective. This approach also allows visualization of the full (long-term) Fe(II) oxidation profiles, which were not even shown in the original figure. I have several comments to further improve the readability of Figure 1. First, the updated insets appear to be downscaled versions of the original plots. As a result, the font size is extremely small and difficult to read. I recommend re-plotting these panels with increased font sizes, symbol sizes, and line thicknesses to enhance clarity. In addition, for the dissolved O₂ and cell count insets, the y-axis ranges extend well beyond what is necessary to highlight differences between conditions. For example, the original panels 1E and 1F span ranges of 1–600 and 0–3 × 10⁸, respectively, whereas adjusting these to 0–200 and 0–5 × 10⁷ would make the differences more apparent. Finally, the revised figure is missing panel labels (A through L), which should be added for clarity and ease of reference.

We appreciate the opinion of the reviewer, and we also agree that consistent x-axes, on the first sight, would appear useful. We therefore visualize the data on the same x-axes (time) in Figure 2 where we have now added the iron oxidation panels over the entire course of the experiment. However, we prefer to keep Figure 1 as it is in the manuscript, because our aim in this figure was to shift the reader's attention to the first days of active Fe(II) oxidation. As we decided to adapt the x-axes to the respective Fe(II) oxidation period, for consistency and comparability, we set the y-axes maxima consistently over all plots in figures 1 and 2 (e.g., all oxygen plots from 0 to 600 μM). We hope that this is also fine with the editor and reviewer. Since we prefer keeping the figure as it is (without the inserts), there is no issue with the smaller font size and therefore we did not change it in Fig. 1.

Despite the revisions made by the authors, I still find the main text very difficult to read. I previously suggested modifying how the results are presented in the section "Effects of different silica concentrations on Fe(II) oxidation by cyanobacterial O₂ under alternating day–night cycles." The authors indicated that they prefer to retain the current structure. While I understand their rationale for organizing the results by Fe(II) condition and then by the presence or absence of silica, the section remains overly verbose and repetitive. As currently structured, the authors sequentially describe each Fe(II) condition, followed by color changes, cell growth, O₂ accumulation, Fe(II) oxidation, and silica precipitation, repeating this format for both silica-free and silica-amended experiments. This results in an entirely descriptive section spanning roughly ten paragraphs. Although I appreciate the authors' effort to be thorough, I feel that the key findings are obscured by the level of detail provided. In addition, most numerical values reported often add noise rather than insight, particularly when the implications of these differences are not explicitly discussed. For example, in lines 134–138, the

reported differences in final cell densities between the “no silica” and “high silica” setups are relatively small, and it is unclear whether these differences are meaningful in the context of the study. Similar examples occur throughout much of the Results section. I therefore recommend restructuring this section by (1) summarizing observations more concisely and (2) presenting key quantitative results in a table, similar to Table 1, instead of listing every single number in the main text. Such a table could report rates of cell growth and dissolved O₂ accumulation, potentially over both short and long timescales (as this timescale difference is relevant for the study). This approach would substantially improve readability while ensuring that the most important results are clearly communicated.

As requested by the reviewer, we have added more geochemical information to Table 1 and now refer to it in the text. We acknowledge that the increased level of detail might be distracting to some readers and as suggested, we therefore have shortened the text slightly. However, we believe that presenting these results systematically provides a clearer overall picture of the experiments. By organizing the data starting from the respective Fe(II) conditions, followed by presenting the results on color changes, cell growth, O₂ accumulation, Fe(II) oxidation, and silica precipitation, we maintain a structured and comprehensive overview of the dataset. We hope the reviewer and editor appreciate this rationale and the authors’ approach to presenting the experimental results.

Along similar lines, the logic of the Discussion is very difficult to follow in its current form. For example, the authors begin by stating that “oxygenic photosynthesis on the early Earth proceeded without harmful ROS stress” before even explaining the postulated mechanisms by which ROS formation is mitigated by the presence of silica or by day–night cycles. The Discussion should be structured as a stepwise logical argument that progresses from specific experimental results to broader implications, which is not the case in its current form. For instance, it would be clearer first it is described the specific mechanisms through which silica and day–night cycling attenuate ROS formation in the authors’ experiments. This could then be followed by a broader discussion of the implications for how oxygenic photosynthesis may have proceeded on the early Earth without ROS toxicity, and how this influenced water-column redox structure and O₂ distribution in seawater.

We agree with the reviewer and therefore have re-organized the discussion accordingly. The new order now is:

- ROS formation and the mechanistic function of Si for ROS prevention
 - Day-night-cycles reduce ROS formation compared to continuous light and static conditions
 - Implications of the oxygen distribution in the seawater
 - Oxygenic photosynthesis on early Earth proceeded without harmful ROS stress to cyanobacteria
- The very conclusive paragraph has been kept at the end.

Minor comments:

1. Line 3.- Would be relevant to cite “Fournier, G. P., Moore, K. R., Rangel, L. T., Payette, J. G., Momper, L., & Bosak, T. (2021). The Archean origin of oxygenic photosynthesis and extant cyanobacterial lineages. *Proceedings of the Royal Society B*, 288(1959), 20210675.” where they also support an Archean origin of total group cyanobacteria.

We added the citation to lines 39-40.

2. Line 74-75. Missing a reference.

We added Kappler et al. (2021) “An evolving view on biogeochemical cycling of iron” as reference.

Lines 76-78: “While the Fe³⁺ drives the formation of Fe(III) minerals, the generated radicals can undergo further reactions, forming additional ROS like ozone (O₃) and hydroperoxyl radicals (OOH•) (for a review see ³³).”

3. Line 130-133. Refer to Figure 1 in this sentence.

Done as suggested.

Lines 131-132: “The results are plotted in **Fehler! Verweisquelle konnte nicht gefunden werden.** on the respective timescale of the active Fe(II) oxidation, [...]”

4. Lines 360-364. This is the motivation and hypothesis of the study, and it was not incorporated effectively in the introduction. I would suggest exchanging these lines (360-364) to the ones in the introduction which are a summary of what is shown here: “Dreher and colleagues reported increasing Fe(II)(aq) toxicity with increasing Fe(II)(aq) concentrations, but observed markedly enhanced bacterial growth compared in the presence of silica.”

As suggested by the reviewer, we have switched the paragraphs accordingly.

Lines 111-115: “Our working hypothesis is that silica binds Fe(II)_{aq}, forming Fe-Si-aggregates that slow Fe(II) oxidation and consequently inhibit ROS formation. Previous studies have shown that reduced Fe(II) oxidation rates lead to lower ROS formation³⁶. Consistent with this, Dreher and colleagues³⁷ showed, using SEM/EDS, that Fe-Si aggregates precipitate under similar experimental conditions, further supporting our hypothesis.”

Lines 447-450: “Dreher and colleagues³⁶ reported increasing Fe(II)_(aq) toxicity with increasing Fe(II)_(aq) concentrations, but observed markedly enhanced bacterial growth compared in the presence of silica. Based on these results, we hypothesize that silica reacts with Fe(II)_(aq), thus distinctly lowering Fe(II) reactivity and preventing Fe(II)-induced ROS formation.”

5. Line 380 .- “Taken together, our lab coupled with the modeling results...” The sentence is missing “experiments” after “lab”.

Changed as suggested.

Response to Reviewer #2:

(Remarks to the Author): I thank the authors for their detailed responses to comments and suggestions. I think that the authors integrated all key suggestions very well and strengthened an already excellent manuscript. I have no additional comments beyond those addressed in my review and those of the other reviewers. The manuscript is well written, the results and interpretations are novel, and the findings contribute significantly to our understanding of interplays between early cyanobacteria and environmental conditions on the early Earth. I support publication of this manuscript.

We thank reviewer #2 for the nice feedback and the effort taken in reviewing our manuscript again.

Response to Reviewer #3:

(Remarks to the Author): I am satisfied with all but one of the authors' responses to my comments. My single objection is a small one (but still important, I think). When I recommended the authors add a few lines of text stressing the importance of high Silica in Archean seawater, I was hoping to see a sentence describing the supporting data for Silica-rich Archean seawater. Archean seawater WAS silica rich; this was the point I wanted them to make clearer. My apologies for the unclear directions.

We would like to thank reviewer #3 for the positive feedback and the clarification. We now added a small paragraph about the evidence for high silica concentrations in the Archean seawater.

Lines 51-57: “Prior to the emergence of silica-precipitating microorganisms (e.g., diatoms), silica inputs from chemical weathering of early continental crust and hydrothermal vents led to elevated dissolved silica concentrations in the seawater. These concentrations were constrained chiefly by saturation with respect to cristobalite or amorphous silica, on the order of 0.67-2.2 mM^{18,19}. Recent studies further suggest that under these ‘high silica’ concentrations, the primary precipitate may have been a gel-like composite of ferric oxyhydroxide and silica²⁰⁻²².”

ROUND 1 REVIEWER 2 ATTACHMENT:

This study seeks to understand the chemical factors that may have influenced the proliferation of cyanobacteria and the Great Oxidation Event, topics of broad interest to the geoscience community. The study specifically focuses on the potential connections among iron toxicity, oxidative stress, and elevated Fe(II) concentrations in Archean oceans as factors that may have limited cyanobacterial proliferation and the impact of high SiO₂ concentrations in Archean seawater as a potential mitigating factor in iron toxicity and ROS generation.

The study uses an experimental approach to test the impact of Fe(II) concentrations, oxidative stress, and SiO₂ concentrations on cyanobacterial growth and O₂ production. The authors measure O₂ concentrations, Fe(II) and SiO₂ concentrations, Fe_{total}, cell numbers, and ROS production to characterize the interactive effects of Fe(II) and SiO₂ on cyanobacterial physiology, oxygen production, and growth. These experimental findings are paired with geochemical modeling to characterize total O₂ production under Archean-type conditions and scale the laboratory results to early oceans. Together, the experimental and modeling results demonstrate how day/night cycles and elevated SiO₂ concentrations in Archean oceans could have mitigated ROS stress and Fe(II) toxicity, playing instrumental roles in the proliferation of cyanobacteria and the GOE.

Overall, the manuscript is well written, the experiments are well designed, and the results are compelling and novel. This study provides important new experimentally derived insights into the ways in which Fe(II) toxicity and mitigation of toxicity and ROS generation by SiO₂ may have impacted cyanobacterial proliferation, oxygen accumulation, and the GOE. The experimental results convincingly demonstrate that Fe(II) toxicity negatively impacts cyanobacterial growth and O₂ production. The experiments also clearly demonstrate that under elevated concentrations of SiO₂ (similar to estimates of Archean oceans), the impact of Fe(II) toxicity and ROS stress is diminished, evidenced by higher growth and O₂ production and diminished ROS in high SiO₂ cultures. Using these experimental findings, the study refines models that predict O₂ accumulation and ocean Fe(II) concentrations during the Archean and demonstrates that elevated SiO₂ concentrations in Archean oceans may have played a key role in cyanobacterial proliferation and the GOE.

The findings are novel, and the results will be of broad interest. I support publication of the manuscript, but I have several comments and suggestions outlined below regarding details of experimental design, presentation of the data, and potential discussion points that would strengthen the manuscript. Broadly, 1) the introduction and discussion could be strengthened by posing a possible mechanism to explain how silica is mitigating the impacts of Fe(II) toxicity and ROS, 2) some details of the experimental design should be mentioned in the main text (see below), 3) the data in figure 1 should be plotted on the same timescales to more robustly show

trends highlighted in the text, 4) there are some mentions of the precipitates but no data characterizing what these precipitates are – if these data were collected, it would be useful to show them. These comments and other minor suggestions are outlined in the attached document with line references.

Detailed suggestions:

- Line 106: The introduction does a great job of painting a picture both of the connection between Fe(II) and oxidative stress, the potential mitigation of these impacts by elevated SiO₂ concentrations, and relationship between Fe and SiO₂ in BIFs indicating iron-silica interactions in Archean oceans. However, the specific connection between Fe(II) stress and SiO₂ mitigation and the potential underlying mechanism is not clear. What is the specific hypothesis or hypothesized mechanism being tested? That Si may remove Fe(II)? Or that it may directly interact with the cells in some way? Mentioning the hypothesized mechanism in the introduction and the discussion (see later comments) would be of interest to the community and strengthen the findings from the study.
- Line 120: Although the experimental design is outlined in the results section, it would be useful to include a brief note at the beginning of the results section to help the reader follow the results. Just a sentence or two noting the basic setup, number of replicates, and variables being tested.
- Results/Methods: Related to this, it is not clear between the initial growth conditions and experimental conditions whether the experiments were conducted under anoxic conditions. It would be helpful to explicitly state this at the beginning of the results section as this is a key parameter. Whether the experiments were conducted under oxic or anoxic conditions has strong implications for Fe oxidation both at time 0 and over the course of the experiments as well as the changes in O₂ observed.
- Results/Methods: Also related to the experimental design, were sterile controls used to confirm that Fe(II) was not oxidized in the absence of cyanobacteria?
- Line 126 and Figure 1A: Why does Fe(II) increase in the control condition by the end of the experiment?
- Figure 1: Why are the plots in Fig 1 plotted on different timescales? It would be more useful to directly compare the experiments over the same timescales (x axis) and then re-scale the Y axes so that small changes (like those in A, E, F, and I) can be more clearly seen.
- Line 148: Although error is stated in the table, it would be helpful to state in the text to make it clear that the oxidation rates at this concentration of silica and Fe(II) were similar within error. As written, it seems to imply that there is a difference in the oxidation rates with and without silica, but these values are similar within error at the 500 μM Fe(II) concentration based on the table.
- Line 163: The color changes do not seem to be the key result, especially given the potential confounding variables and mechanisms underpinning the color changes that the

authors note. These "color change" results could be moved to the supplemental document so that the results of the main text could more clearly focus on the key quantifiable trends.

- Line 182: Is Fe still getting oxidized in the high silica experiments? If so, why were Fe(III) precipitates (i.e., orange colors) not observed? Or is the Fe(II) being complexed by the silica? Do you have any data (e.g., SEM/EDS) characterizing the nature of the precipitates? This could be helpful in understanding the underlying mechanism by which silica may be buffering against Fe(II) toxicity.
- Line 189: Do you have a hypothesis to explain this variability?
- Line 231: Why was a different starting concentration of silica used in these experiments compared to the growth experiments (1600 μM compared to 2200 μM)?
- Line 363: Here, as in the introduction, you could consider posing a potential mechanism to explain *how* silica mitigates the impact of Fe(II) toxicity and ROS. The results of the study are very compelling and demonstrate that there is a correlation and that SiO_2 may have been a key component to Fe (II) toxicity and ROS mitigation, but these results would be even more impactful if you outlined some potential mechanisms.
- Line 414: Do you think that the length of the day matters?
- Line 458: Do you have any data aside from observations of cloudiness in the bottles to characterize the precipitates?